# Damped circadian oscillation in the absence of KaiA in *Synechococcus*

Naohiro Kawamoto[1], Hiroshi Ito[2], Isao T. Tokuda[3] & Hideo Iwasaki[1✉]

Proteins KaiA, KaiB and KaiC constitute a biochemical circadian oscillator in the cyanobacterium *Synechococcus elongatus*. It has been reported *kaiA* inactivation completely abolishes circadian oscillations. However, we show here that *kaiBC* promoter activity exhibits a damped, low-amplitude oscillation with a period of approximately 24 h in *kaiA*-inactivated strains. The damped rhythm resonates with external cycles with a period of 24–26 h, indicating that its natural frequency is similar to that of the circadian clock. Double-mutation experiments reveal that *kaiC*, *kaiB*, and *sasA* (encoding a KaiC-binding histidine kinase) are all required for the damped oscillation. Further analysis suggests that the *kaiA*-less damped transcriptional rhythm requires KaiB-KaiC complex formation and the transcription-translation feedback loop, but not the KaiC phosphorylation cycle. Our results provide insights into mechanisms that could potentially underlie the diurnal/circadian behaviors observed in other bacterial species that possess *kaiB* and *kaiC* homologues but lack a *kaiA* homologue.

[1] Department of Electrical Engineering and Biological Science, Waseda University, Tokyo 162-0056, Japan. [2] Faculty of Design, Kyushu University, Fukuoka 815-8540, Japan. [3] Graduate School of Science and Engineering, Ritsumeikan University, Shiga 525–8577, Japan. ✉email: hideo-iwasaki@waseda.jp

Many organisms are exposed to changes in environmental cycles due to day–night alternation. The circadian clock is considered a mechanism for adapting to such a cyclic environment, and is conserved across species ranging from bacteria to higher plants and animals. The core components of the cyanobacterial circadian system are the KaiA, KaiB, and KaiC proteins, as studied in *Synechococcus elongatus* PCC 7942[1,2]. KaiA elevates the autophosphorylation activity of KaiC[3], and KaiB inhibits the effect of KaiA[4,5]. The resulting KaiC phosphorylation cycle with a period of 24 h is sustained even in the absence of the transcription–translation feedback loop (TTFL) process[6], and it can be reconstituted in vitro[2]. The Kai-based post-translational oscillator (PTO) controls genome-wide transcription/translation profiles and various physiological phenomena in a circadian manner[7,8].

In *Synechococcus*, inactivation of *kaiA* dramatically reduces the magnitude of both KaiC phosphorylation and *kaiBC* expression[1]. Thus, KaiA has been reported repetitively as an essential clock component in the cyanobacterial circadian system. Interestingly, the *kaiB* and *kaiC* genes are found not only in cyanobacteria but also in other proteobacteria and Archaea, while *kaiA* is only found in cyanobacteria. Detailed phylogenic analysis by Dvornyk and colleagues (2003) suggested that *kaiA* is evolutionarily the youngest among the three genes[9]. Some marine cyanobacterial species, such as *Prochlorococcus marinus* MED4 and PCC 9511, are known to lack *kaiA*. It has been proposed that in these species, *kaiA* gene was lost after evolution of the intact *kaiABC* system[10]. Consistent with the proposed role of KaiA, *kaiA*-lacking species fail to exhibit oscillation under continuous conditions[11,12], whereas they display diurnal variations in transcription and cell-cycle control under light–dark (LD) cycles.[13–15] Moreover, the diurnal but not free-running rhythm in nitrogen fixation has been reported in even non-cyanobacterial purple bacterium, *Rhodopseudomonas palustris*, which harbors *kaiB* and *kaiC* homologs without *kaiA*[16]. Differing from KaiC in *Synechococcus*, KaiA is not essential for enhancing the basal autophosphorylation activity of the KaiC homolog in *Prochlorococcus* MED4 and *Rhodopseudomonas*[12,16]. In both species, KaiC homologs undergo phosphorylation in the light and dephosphorylation in the dark[16,17]. Based on these results, prior research discussed the possibility of a non-self-sustaining timing system in cyanobacterial and purple bacterial species lacking *kaiA*[11,12,16–18]. More recently, a comparative bioinformatics study by Schmelling et al.[19] reported a possible conserved gene network composed of *kaiB*, *kaiC*, *sasA*, *rpaA*, *rpaB*, *ldpA*, *cpmA* and, *ircA* among cyanobacterial species, and proposed these genes as possible components of the prototypic hourglass-like timing system. There are a couple of possible mechanisms for the timing system other than a self-sustained oscillator from a mathematical viewpoint. One mechanism is the hourglass model, which can respond to periodical environments, but does not exhibit any oscillations under constant conditions. The other possibility is damped oscillation, which can display oscillations under constant conditions, although its amplitude can decay exponentially. In both *Prochlorococcus* and *Rhodopseudomonas*, these possibilities have not been experimentally validated.

On the other hand, we recently demonstrated that the reconstituted self-sustained KaiC phosphorylation cycle in vitro is turned to exhibit damped oscillation under low temperatures, following the Hopf bifurcation property[20]. At temperatures of less than 19 °C, KaiC phosphorylation displays damped oscillation with a period of approximately 24 h and gradual loss of amplitude. Importantly, resonant oscillation regained when temperature pulses were periodically applied with a period of a circadian range, undoubtedly demonstrating the presence of a damped oscillatory mechanism.

In the present study, we demonstrate that *Synechococcus* exhibits KaiA-independent damped oscillation in a transcriptional output, which is resonated with external cycles with a period of a circadian cycle. We further investigate the possible involvement of complex formation between KaiB and KaiC and the TTFL process in generating damped oscillation in the absence of KaiA.

## Results and discussion

### Damped oscillation in *kaiBC* promoter activity in the absence of KaiA.

It has been reported that inactivation of *kaiA* abolishes transcriptional rhythms as monitored by a bioluminescence reporter[1], which was the primary experiment insisting that KaiA is essential for driving circadian rhythms. However, careful re-examination of the bioluminescence profile of *kaiBC* promoter ($P_{kaiBC}$) activity led us to uncover that the bioluminescence level exhibited damped oscillation in a *kaiA*-inactivated ($kaiA^-$) *Synechococcus* strain under continuous light (LL) conditions after two LD cycles (Fig. 1a). The average level of $P_{kaiBC}$ activity in the

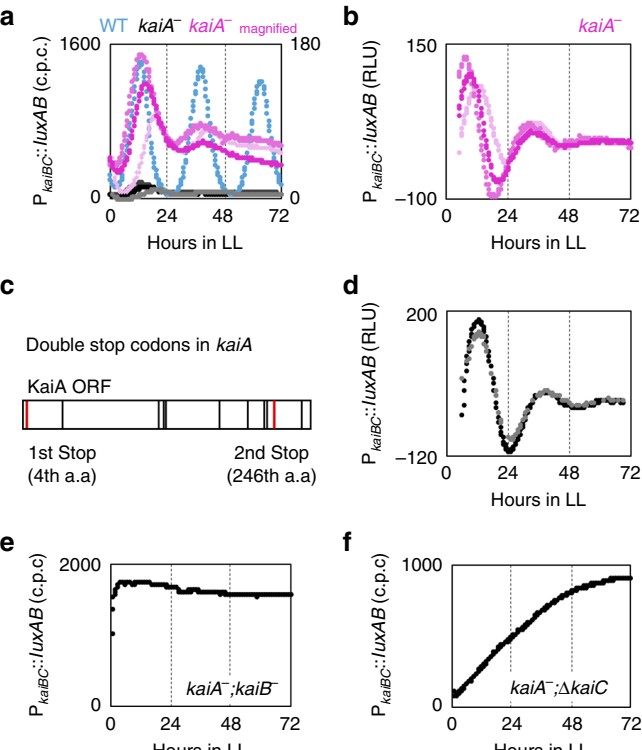

**Fig. 1 kaiA-inactivated strain exhibits damped oscillation of kaiBC promoter activity in a KaiB- and KaiC-dependent manner. a** Bioluminescent profiles of the wild-type (WT, blue) and $kaiA^-$ (black and gray; magenta for magnified scale, $n = 3$) strains under continuous light (LL). Bioluminescence is presented as counts per colony (c.p.c.). **b** Detrended bioluminescence profiles of damped bioluminescence rhythms in $kaiA^-$ strains. Trends of bioluminescence profiles shown in (**a**) were removed with *lag* ($=10$ h). For detrended bioluminescence values, we used relative light units (RLU). It should be noted that phase information is not available for detrended data because the moving average method generates a delay. **c** Two red lines indicate two nonsense (stop codon) mutations introduced into the 4th and 246th codons in *kaiA*. Nine other ATG codons are shown in black lines. Note that the original KaiA protein consists of 284 amino acid residues, and the translational start codon of *kaiA* is GTG instead of ATG. **d** Detrended bioluminescence profiles of damped bioluminescence rhythms in the $kaiA^-$ strain with double stop codons. **e**, **f** Bioluminescence profiles of the $kaiA^-$; $kaiB^-$ (**e**) and $kaiA^-$; $\Delta kaiC$ (**f**) strains exhibiting arrhythmia. Source data are provided as a Source data file.

$kaiA^-$ strains was similar to the trough level of that in the wild-type (WT) strain, as reported in previous studies[3,21]. The bioluminescence level in the $kaiA^-$ strain peaked around hour 16 under LL conditions. When the basal trend of the profile was removed, two or three peaks were evident at regular intervals of approximately 24 h (Fig. 1b). Using the model fitting by Westermark et al.[22], the time constant of amplitude decay is calculated to be ~12 h, which means that the oscillation amplitude diminishes to ~20% compared with the previous cycle. It should be noted that in previous studies, at least one[23,24] or two cycles[3] of $P_{kaiBC}$ bioluminescence were observed retrospectively, although they were considered arrhythmia at that time because the amplitude of the damped oscillation in $kaiA^-$ strains was extremely low compared with that of the WT strain. In these studies, a partial segment of the upstream region of the $kaiBC$ gene (previously named D4)[21] has been used as the $kaiBC$ promoter to drive bioluminescence because of its highly expressing level. The selection of this promoter unit might be beneficial to detect the damped oscillation profile with lower expression levels due to the lack of $kaiA$. In addition, light intensity seems also important to detect the damped oscillation, since at lower light intensity (15 μmol photon $m^{-2}$ $s^{-1}$), the bioluminescence rhythm was more rapidly damped without showing the second peak of the rhythm (Supplementary Fig. 1).

The $kaiA^-$ strain we used harbors a nonsense mutation at the fourth codon that inactivates $kaiA$[1]. As previously examined[3,25], we confirmed that KaiA was not expressed in the $kaiA^-$ strain (Supplementary Fig. 2). However, the possibility that a truncated form of KaiA is expressed at levels below the detection limit remains. Therefore, another $kaiA^-$ strain in which the 4th and 246th codons were substituted with stop codons was constructed (Fig. 1c). This strain also exhibited the same damped oscillation as the original $kaiA^-$ strain (Fig. 1d). When a $kaiA$-null mutant was additionally mutated to inactivate either $kaiB$ or $kaiC$, the damped oscillation in $P_{kaiBC}$ activity was abolished (Fig. 1e, f). The result confirmed that the low-amplitude damped oscillation was not an artifact. The finding also indicated that the $kaiA$-less damped oscillation requires the function of the KaiB and KaiC proteins.

**Phase resetting and temperature compensation property of the $kaiA$-less damped oscillation.** Three criteria are widely adopted as characteristics of circadian rhythms: (i) free-running rhythm of a period of 24 h under continuous conditions, (ii) entrainability to environmental cycles such as LD or temperature cycles, and (iii) temperature compensation of the period length. To what extent are these circadian characteristics of the canonical clock observed in the $kaiA$-less damped oscillation?

To examine the light-resetting property, we investigated the phase-response curves (PRCs) of the bioluminescence rhythms in the WT and $kaiA^-$ strains against dark pulses of 2, 4, or 6 h at hours 4, 8, 12, 16, 20, 24, or 28 in LL, following two LD cycles to entrain the clock. Under our experimental conditions, dark pulses of 2 or 4 h gave rise to slight phase shifts in the WT strain, whereas dark pulses of 6 h advanced the phase during the subjective day (hour 8 in LL, Fig. 2a), as reported previously[26]. Conversely, the phases of bioluminescence rhythms in $kaiA^-$ strains were much more dramatically shifted against dark pulses, especially those of 4 or 6 h (Fig. 2b). These results suggest that the $kaiA^-$ strain is more sensitive to dark stimuli, and it must be easier to set by LD cycles. The negative correlation between the amplitude of the rhythm and the magnitude of phase shifting has been reported in circadian systems in mammals[27], *Arabidopsis*[28], and the $kaiC^{EE}$ mutant of *Synechococcus*[26]. These phenomena have been interpreted partly by a simple schematic model. Self-sustained circadian clocks have

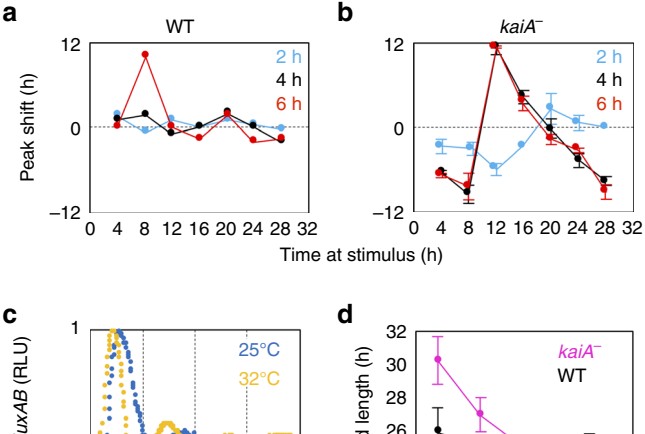

**Fig. 2 Light response and temperature compensation properties of the KaiA-less damped oscillation. a, b** The time of day-dependent phase-response curves of bioluminescence rhythms in the wild-type (WT, (**a**) and $kaiA^-$ (**b**) strains against 2- (blue), 4- (black), or 6-h (red) dark pulses ($n = 3$). Note that the $kaiA^-$ strain exhibited enhanced phase-shifting properties. Dot plots and error bars indicate mean and s.d., respectively. **c** Representative detrended bioluminescence profiles in the $kaiA^-$ strain at 25 or 32 °C. **d** Period length of the sustained rhythm in the WT strain ($n = 3$) and the time difference between the first and second peaks in the $kaiA^-$ strain ($n = 8$) at each temperature. Dot plots and error bars indicate mean and s.d., respectively. Source data are provided as a Source Data file.

been considered to be limit cycle oscillators. In limit cycle theory, a reduction in the amplitude of the self-sustained oscillator is visualized as a limit cycle with a smaller diameter. A stimulus that causes an equivalent change in state variables would give rise to larger phase shifts if the trajectory of the damped oscillation is much smaller than the diameter of a high-amplitude limit cycle on the phase diagram. By contrast, the same stimulus could cause smaller phase shifts with a limit cycle with a larger diameter[29]. Thus, the difference in PRCs between the WT and $kaiA^-$ strains could be discussed similarly: the WT and $kaiA^-$ oscillations would be considered to be a self-sustained limit cycle with larger amplitude and the damped oscillator with smaller amplitude, respectively. An alternative possibility is the enhancement of photic input pathways in $kaiA^-$ strains. For example, disruption of glucose-1-phosphate adenylyltransferase gene ($glgC$) magnifies the dark-induced phase shifting through metabolic changes with a rapid fall of ATP/(ATP + ADP) energy charge in dark conditions, while the amplitude of $P_{kaiBC}$ rhythm is less altered[30]. Although we cannot exclude this possibility, the former model appears more plausible because it is evident that the core oscillatory mechanism must be much more fragile in the absence of KaiA compared with the findings for the canonical circadian pacemaker in the WT strain.

We compared three methods for estimating periods from bioluminescence signals: (i) damped oscillator model fitting by the method of Westermark et al.[22] (Supplementary Fig. 3), (ii) autocorrelation function analysis, and (iii) peak-to-peak interval. For the WT data, the averaged period was estimated to be 25.1 h by all three methods. The averaged period of $kaiA^-$, on the other hand, was estimated to be 25.8, 24.8, and 24.0 h by the model fitting, the autocorrelation, and the first peak-to-peak interval, respectively (Supplementary Table 2). To verify precision of the period estimates, the three methods were applied to artificial data

sets generated from linear damped oscillators, the periods of which were known a priori. As described in detail in supplementary information, the estimation error is given by deviation of the estimated periods from the true ones. As the damping rate was increased, the estimation error increased monotonously for all three methods (Supplementary Fig. 4a–c). This is because the period information is lost quickly by a strong damping. In the model fitting, the estimation error increased to 1 h at a damping rate of $\lambda = 0.05$ and reached to 2 h at $\lambda = 0.1$ (Supplementary Fig. 4a). The autocorrelation analysis gave even larger errors (Supplementary Fig. 4b). The peak-to-peak interval, on the other hand, produced results comparable to those of the model fitting, when the initial two peak-to-peak intervals were averaged as the period estimate (Supplementary Fig. 4c, d). When only the first peak-to-peak interval was used, the estimation errors became much smaller than the model fitting, especially for a damping rate between 0.025 and 0.15 (Supplementary Fig. 4e, f). In noisy damped data, signals are attenuated quickly, while the noise effect becomes non-negligible. This lowers the signal-to-noise ratio over time. For this reason, the first peak-to-peak interval provides the most reliable period information than the other two methods that average long-term properties of the attenuated signals. For a precise estimation of period from damped oscillators, it is more advantageous to utilize the portion, in which the signal is least attenuated. According to our estimate, the damping rate of $kaiA^-$ was approximately 0.05, i.e., within the range where periods are most precisely estimated by the first peak-to-peak interval. Therefore, in the following period analysis, the time intervals between the first and second peaks are regarded as the period.

Next, to examine the temperature dependency of the period of $kaiA$-less oscillation, we compared the period lengths of bioluminescence rhythms between the WT and $kaiA^-$ strains at temperatures of 25, 27, 30, and 32 °C. As shown in Figs. 2c, d, the period length was more sensitive to temperature changes in the $kaiA^-$ strain than in the WT strain. The period lengths in the $kaiA^-$ strain were 30.2 h at 25 °C, 27.0 h at 27 °C, 24.0 h at 30 °C, and 25.0 h at 32 °C, whereas those in the WT strain at these temperatures were 25.9, 25.1, 25.1 and 25.5, respectively. The bioluminescence of $P_{kaiBC}::luxAB$ in $kaiA^-$ has a tendency to decrease at higher temperature. At 35 °C, the oscillation in the $kaiA^-$ became much more dampened than that at 32 °C. The average level of bioluminescence at 35 °C was less than 10% of that at 30 °C, and the second peaks of bioluminescence were not reproducibly observed (Supplementary Fig. 5). The tendency to lower the bioluminescence level at relatively higher temperature has been observed in WT, i.e., the peak value at 38 °C decreased by about 30% of that at 30 °C[31]. At temperatures of 25–30 °C, the calculated $Q_{10}$ value for the period of the rhythm in the WT strain was 1.06 versus 1.59 for the $kaiA^-$ strain. These results indicated that the period length of the $kaiA$-less oscillator is less temperature-compensated than that of the intact KaiABC-based circadian pacemaker. Thus, the damped oscillator in the absence of KaiA is more easily disturbed by alteration of the temperature and light environment than the canonical circadian clock.

**Resonance of damped oscillation in response to external cues**. The damped oscillator could simulate a pendulum with friction. When a pendulum with friction is kicked out, it swings for a while. Importantly, it continues swinging when subjected to forced external stimuli with an appropriate period (or natural oscillation frequency). This property is called resonance. In the KaiABC-based oscillator, we demonstrated that the in vitro-reconstituted circadian oscillator monitored by the KaiC phosphorylation cycle switched to a damped oscillator at low temperatures, and that the amplitude of the damped oscillation is resonated with forced

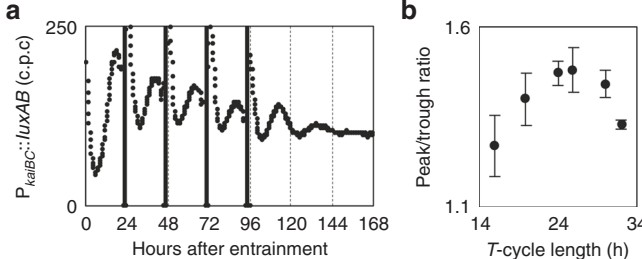

**Fig. 3 Resonance of the kaiA-less damped oscillation in response to cyclic external cues. a** After two night–day cycles to entrain the oscillator, the bioluminescence profile in the $kaiA^-$ strain was monitored under continuous light, whereas the cells were exposed to four cycles of 2-h dark pulses (black bars) with a period of 24 h. **b** After external cues (2-h dark pulses) were given for a period ranging from 16 to 32 h, the peak-to-trough ratio of the bioluminescence profile (as an index of amplitude) was calculated ($n = 3$). Dot plots and error bars indicate mean and s.d., respectively. Source data are provided as a Source Data file.

temperature cycles within a circadian range[20]. We applied four 2-h dark pulses cyclically to the $kaiA^-$ strain under LL at 30 °C with a different external period ($T$) ranging from 16 to 32 h (Fig. 3). The peak-to-trough ratio of the bioluminescence rhythms within the periodic cues was calculated as an index of the amplitude of the forced oscillation. Note that immediately after the dark pulses, transient increases in bioluminescence levels are often observed, as shown in Fig. 3a and Supplementary Fig. 6. This transiently elevated bioluminescence level was not adopted as a peak value of oscillation. Instead, the subsequent trough level and the next peak of the rhythm were compared. As shown in Fig. 3b, when the amplitude of the bioluminescence rhythm was measured as a function of $T$, the amplitude of the forced oscillation peaked at $T$ of approximately 24 and 26. The enhancement of amplitude depending on the period of the external cycles indicated that the damped $kaiA$-less oscillator exhibited resonance, and its natural oscillation frequency at 30 °C is similar to that of the circadian pacemaker.

**Involvement of transcriptional–translational feedback of $kaiBC$ in the damped oscillation**. As mentioned previously, the $kaiA$-less damped oscillation requires the KaiB and KaiC proteins (Fig. 1e, f). To reveal whether other known clock-associated factors are involved in generating the damped oscillation, we introduced mutations into the $kaiA^-$ strain to additionally inactivate $sasA$, $cikA$, or $labA$. SasA is a KaiC-binding histidine kinase that stimulates the phosphorylation activity of RpaA, a master regulator of circadian transcriptional output[8,25,32]. CikA is another histidine kinase that binds to KaiB–KaiC complexes and dephosphorylates RpaA[33]. LabA antagonizes the expression of $kaiBC$ through RpaA[34]. Thus, these three factors are involved in cyclic modulation of the phosphorylation state of RpaA to drive circadian oscillation of clock-controlled genes including the $kaiBC$ operon[35]. As shown in Fig. 4a, when both $kaiA$ and $sasA$ were inactivated ($kaiA^-;sasA^-$), the bioluminescence level ($P_{kaiBC}$ activity) was severely lowered, and the rhythm was not confirmed. $kaiA$-mutant strains, in which either $cikA$ or $labA$ was also disrupted, exhibited increased bioluminescence (Fig. 4b, c) comparable to the peak level in the WT strain (Fig. 1a). The $kaiA$-less damped oscillation was intact when $labA$ was additionally inactivated. When $cikA$ was additionally disrupted, the amplitude of the bioluminescence rhythm was more strongly reduced, although the first peak was still detected (Fig. 4b). Thus, KaiB, KaiC, and the SasA–RpaA two-component system are likely essential for driving $kaiA$-less damped oscillation, and CikA might be also involved in its modulation.

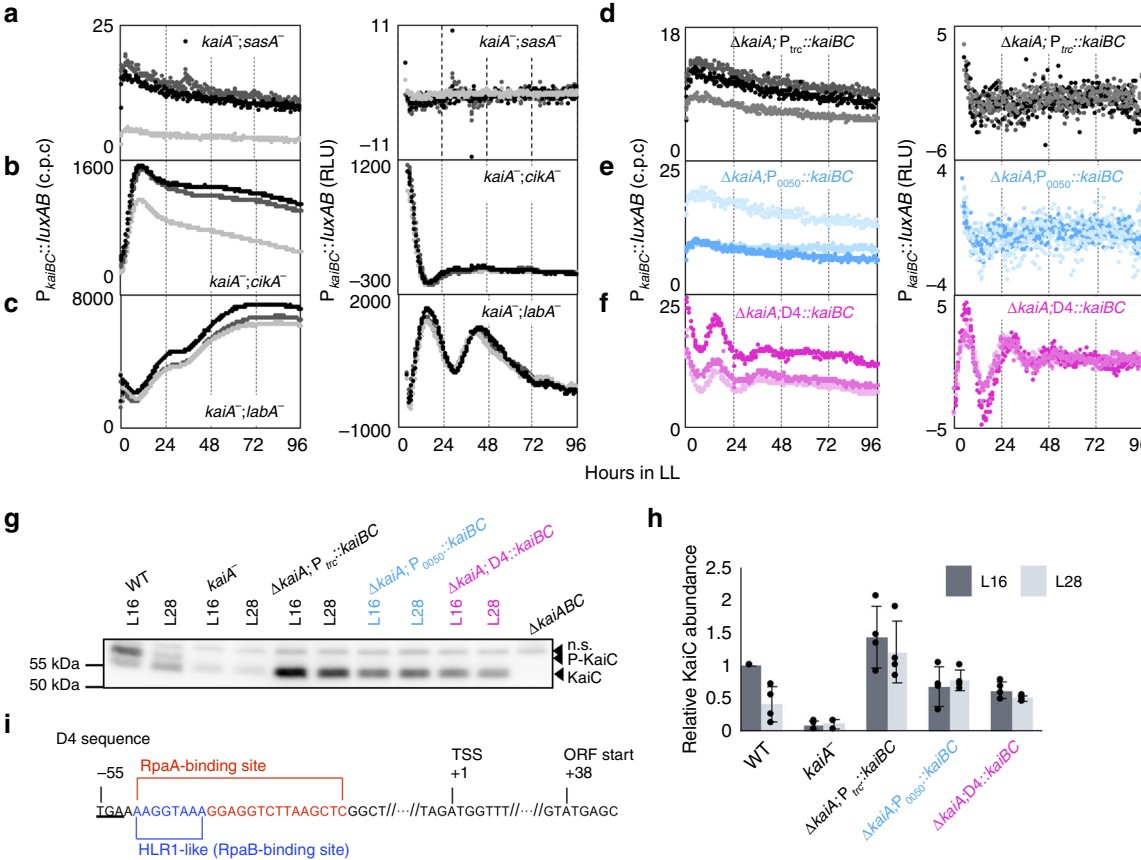

**Fig. 4 Transcriptional–translational feedback of kaiBC is involved in driving the KaiA-less damped oscillation. a–c** The bioluminescence profiles of double mutations of *kaiA* together with *sasA*, *cikA*, or *labA* were measured under continuous light (LL, *n* = 3) (left panels). The detrended profiles of the raw bioluminescence profiles (left) are shown on the right. **d–f** The $P_{trc}$::*kaiBC*, $P_{0050}$::*kaiBC*, or D4::*kaiBC* cassette was introduced into the $\Delta kaiABC$ strain ($\Delta kaiA$; $P_{trc}$::*kaiBC*, $\Delta kaiA$;$P_{0050}$::*kaiBC*, and $\Delta kaiA$;D4::*kaiBC*). The bioluminescence profiles of these strains under LL are shown on the left, and the detrended profiles are presented on the right. **g** Accumulation of KaiC in wild-type (WT) and *kaiA*-null mutant strains as measured using Western blotting. The upper and lower bands for KaiC denote the phosphorylated (P-KaiC) or hypophosphorylated (KaiC) forms, respectively. A band shown with nb indicates a nonspecific signal. **h** Densitometric analysis of (**g**) (WT, $\Delta kaiA$;$P_{trc}$::*kaiBC*, $\Delta kaiA$;$P_{0050}$::*kaiBC*, and $\Delta kaiA$;D4::*kaiBC*, *n* = 4; *kaiA*⁻, *n* = 3). The data were normalized to the signal in WT at LL16 shown in (**g**) of 1.0. Black and gray bars indicate mean of the data after 16 (L16) and 28 h (L28) of light exposure. Dot plots represent the density of individual bands, and error bars indicate s.d. **i** The truncated *kaiBC* (D4) promoter consists of a 92-bp sequence upstream of the *kaiB* open-reading frame (ORF), including an RpaA-binding site and a HLR1-like RpaB-binding site. The D4 sequence started with the stop codon of the *kaiA* ORF (underlined). Source data are provided as a Source Data file.

These results could be interpreted in light of the possible involvement of the TTFL process in $P_{kaiBC}$ activity through output factors (especially SasA), which are modulated by the KaiB and KaiC clock proteins to drive the damped oscillation in the absence of KaiA. To test this possibility, we tried to constitutively express *kaiBC* under the control of the *trc* promoter instead of the endogenous promoter. When *kaiBC* was induced in the *kaiBC*-knockout mutant strain from a targeting site, namely neutral site II (NSII), in the presence of the inducer IPTG (5 μM), circadian bioluminescence rhythms were recovered[36]. By contrast, when *kaiBC* was induced in the *kaiABC*-depleted background, the bioluminescence level was suppressed at lower levels, and no rhythmicity was detected (Fig. 4d). However, the KaiC expression level of this strain in the absence of IPTG already exceeded that of the original *kaiA*⁻ strain (Fig. 4g, h). To exclude the possibility that the excess amount of KaiC suppressed the damped oscillation, we induced *kaiBC* under the control of a 700-bp sequence upstream of the translation start of *synpcc*7942_0050 from NSIII in the *kaiABC*-null background ($\Delta kaiA$;$P_{0050}$::*kaiBC* strain). This segment was used previously to constitutively express the *kaiBC* operon in a *kaiBC*-depleted strain, which exhibited unstable (less synchronized) circadian transcription/

translation rhythms[37]. The *kaiBC* expression level is maintained in this strain at the mean level of that in the WT strain in the presence of KaiA[37]. Consistently, the KaiC expression level in the $\Delta kaiA$;$P_{0050}$::*kaiBC* strain was also equivalent to that of the WT strain and lower than that of the $\Delta kaiA$;$P_{trc}$::*kaiBC* strain (Fig. 4h). As shown in Fig. 4e, the resulting strain also failed to display any damped oscillation in the absence of *kaiA*. Thus, *kaiA*-less damped oscillation would require the TTFL associated with *kaiBC* expression.

To further examine this possibility, we attempted to induce *kaiBC* under the control of a truncated *kaiBC* promoter (D4) that retained an RpaA-binding site[8,21] in the *kaiABC*-depleted cells ($\Delta kaiA$;D4::*kaiBC* strain). The RpaA-binding site confers primary *cis*-targets of RpaA, which is harbored by most high-amplitude clock-controlled genes[21]. For rhythmic expression of the *kaiBC* operon, this binding site on the *kaiBC* promoter is proposed to be a key element with which the TTFL is fulfilled. Kutsuna et al. demonstrated that the D4 sequence is sufficient to permit negative and positive feedback from KaiC and KaiA, respectively, and it is interpreted by the presence of the RpaA- and RpaB-binding sites in the promoter[8,38] (Fig. 4i). As shown in Fig. 4f, although the $P_{kaiBC}$ bioluminescence level was severely reduced, at

least one cycle of damped oscillation was reproducibly observed. The lower bioluminescence (*kaiBC* promoter activity) is most likely due to a negative feedback effect from overproduced KaiB and KaiC proteins under the control of D4 promoter activity. It has been demonstrated that D4 promoter activity is highly elevated compared with that of the intact *kaiBC* promoter, which overlaps with the *kaiA* open-reading frame (ORF)[21]. Indeed, we confirmed that the Δ*kaiA*;D4::*kaiBC* strain accumulated more KaiC protein than the *kaiA* single mutant (Fig. 4h). More importantly, the KaiC level in the Δ*kaiA*;D4::*kaiBC* strain was equivalent to that in the Δ*kaiA*;P_{0050}::*kaiBC* strain (Fig. 4h). The KaiC phosphorylation levels in the three strains in the absence of KaiA were much lower than those of the WT strain, as expected (Supplementary Fig. 7). Thus, the failure to drive damped oscillation in the Δ*kaiA*;P_{0050}::*kaiBC* strain is not attributable to excessive KaiC induction compared with that in the *kaiA*⁻ strain, but it is instead due to the lack of the RpaA-based TTFL process. It should be noted that the level of KaiC accumulation is higher at subjective night (hour 16 in LL) than at subjective morning (hour 28) in the WT strain because of the TTFL process (Fig. 4h). Although it was not statistically significant, we observed a similar tendency in Δ*kaiA*;D4::*kaiBC* cells. Thus, the generation

of *kaiA*-less damped oscillation requires, at least in part, the RpaA-mediated TTFL process, although the detailed mechanism remains unknown.

**KaiB–KaiC complex formation for determining the period length of the *kaiA*-less damped oscillator.** In the absence of KaiA, the level of phosphorylated KaiC is dramatically reduced in *Synechococcus*[3]. In this study, to verify whether any specific phosphorylation state of KaiC is required for the damped oscillation, we introduced mutations into *kaiC* to constitutively mimic either the hypophosphorylated or hyperphosphorylated form of KaiC in *kaiA*⁻ strains. When *kaiA* is intact, substitution of KaiC phosphorylation sites (S431 and T432) with phosphomimic glutamate residues (*kaiC*^{EE}) does not abolish transcriptional rhythms, but it sustains rhythms with a lengthened period of 48 h[26,31]. When *kaiA* was inactivated in the *kaiC*^{EE} background, surprisingly, the period length of the damped bioluminescence rhythm was less affected and even shortened by approximately 3 h (Fig. 5a). Thus, the KaiC phosphorylation cycle is not necessary for the damped oscillation. On the contrary, substitution of the KaiC phosphorylation sites with alanine residues (*kaiC*^{AA}) to mimic the hypophosphorylated form abolished any rhythmicity in the absence of

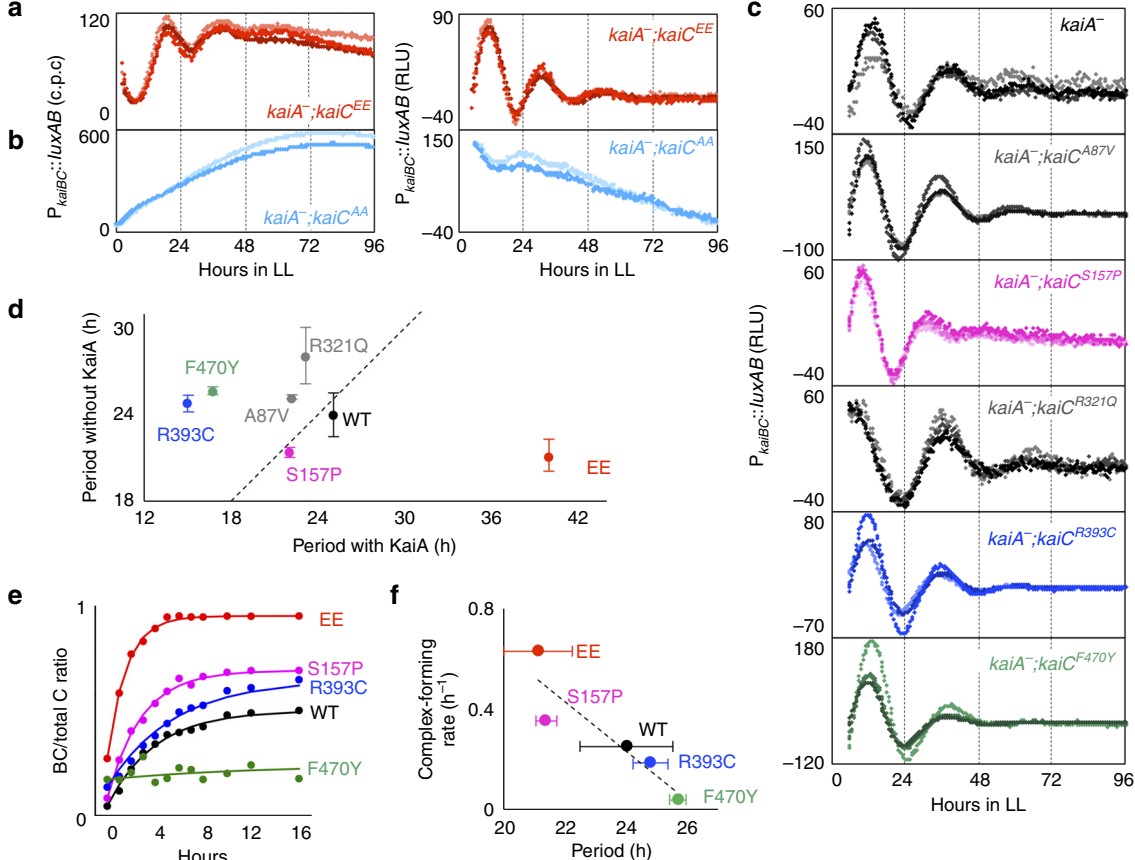

**Fig. 5 KaiB–KaiC complex formation as a key process for generating the kaiA-less damped oscillation. a, b** Bioluminescence profiles of double mutants featuring *kaiA*⁻ together with *kaiC*^{EE} (hyperphosphomimic) (*n* = 3) or *kaiC*^{AA} (hypophosphomimic) (*n* = 2) under LL (left) and the detrended data (right). **c** The bioluminescence profiles of double mutants in which each already-known periodic mutation of *kaiC* was introduced into the *kaiA*-null background (*n* = 3). **d** The intervals between the first and second peaks of the dampened bioluminescence rhythm in the double mutants were plotted against the original period length of the corresponding *kaiC* mutation in the presence of *kaiA*. The dashed line indicates the 1:1 line (*kaiA*-null strains, WT, and *kaiC*^{EE}, *n* = 3; other strains, *n* = 1). Dot plots and error bars indicate mean and s.d., respectively. **e** KaiB–KaiC complex formation profiles analyzed via native PAGE (*n* = 2). Dot plots present the mean values, and the curved lines present data fitted using the least-squares method. **f** The calculated KaiB–KaiC complex formation rates (*k* value described in "Methods") are plotted against the period lengths of the *kaiA*-less damped oscillator in the double-mutant strains with each indicated *kaiC* mutation. Dot plots and error bars indicate mean and s.d., respectively. Correlation coefficient is 0.91. Source data are provided as a Source data file.

KaiA (Fig. 5b), as reported for intact KaiA[39]. The result that the KaiC$^{AA}$ mutant protein fails to show damped oscillation is interesting, because in the original *kaiA*$^-$ strain, the (WT) KaiC is also hypophosphorylated but still able to produce the damped oscillation. Although the majority of the KaiC protein is dephosphorylated in *kaiA*-null mutants, we cannot eliminate the possibility that some KaiC molecules undergo KaiA-independent autophosphorylation, which is important for driving the damped oscillation, whereas it was difficult under our experimental conditions to detect residual phosphorylated KaiC via Western blotting (Figs. 4g and S7). In addition, because of the higher ATPase activity of KaiC$^{AA}$ than intact KaiC and KaiC$^{EE}$, KaiC$^{AA}$ almost does not bind to KaiB in vitro[40]. Alternatively, we cannot eliminate the possibility that the KaiC$^{AA}$ mutation is not a simple mimic of the unphosphorylated form of KaiC, and may have additional abnormalities in its biochemical property: its abnormally higher ATPase may manifest this side effect.

It has been demonstrated that KaiB–KaiC complex formation is dependent on the phosphorylation state of KaiC.[40–46] For example, the recombinant KaiB protein associates with the WT or hyperphosphomimic KaiC protein in the absence of KaiA in vitro, whereas it associates much less with KaiC$^{AA}$[44,46]. Thus, complex formation between KaiB and KaiC would be important for generating the *kaiA*-less damped oscillation.

Before addressing this hypothesis, we examined the effects of *kaiC* mutations (A87V, S157P, R321Q, R393C, and F470Y) on the damped oscillation in *kaiA*$^-$ strains. These mutations are reported to change the period length of the reconstituted KaiC phosphorylation cycles or bioluminescence rhythms in *Synechococcus*[2,47,48]. The period lengths of P$_{kaiBC}$ bioluminescence rhythms in the presence of KaiA with KaiC variants A87V, S157P, R321Q, R393C, and F470Y in LL were 22.2, 22.1, 23.2, 15.1, and 16.8 h, respectively, under our experimental conditions. When *kaiA* was inactivated, the period lengths of damped oscillation in these variants were 25.2, 21.4, 28.1, 24.8, and 25.7 h, respectively (Fig. 5c, d). Only when the S157P mutation was introduced was the tendency of period variation in the presence or absence of KaiA consistent. The damping rates were 0.083, 0.069, and 0.091 for S157P, F470Y, and EE mutations in *kaiA*$^-$ background, respectively, and the amplitude was quickly reduced to about 10–15% per each period. On the other hand, interestingly, the damping rate of *kaiA*$^-$;*kaiC*$^{R321Q}$ was lower (0.027), and the amplitude remained about 40% after one cycle of oscillation.

If KaiB–KaiC complex formation is important for determining the period length of the damped oscillation, the rate of KaiB–KaiC association with different KaiC mutations in the absence of KaiA would be linked with variation in the period length. To test this possibility, we examined the time course of KaiB–KaiC complex formation after incubating recombinant proteins in vitro via native polyacrylamide gel electrophoresis (Figs. 5e and S8). When the rate of KaiB–KaiC complex formation was compared, we found a striking negative correlation between the initial rate and the period length of the damped oscillation (Fig. 5f); specifically, faster KaiB–KaiC complex formation was associated with shorter period lengths. For example, the *kaiC*$^{EE}$ mutant, which had the shortest period in the absence of KaiA, accelerated KaiB–KaiC complex formation, whereas *kaiC*$^{F470Y}$, which had the longest period, slowed complex formation. As expected, in the presence of KaiA, such a simple correlation between KaiB–KaiC complex formation and the period length was not observed (Supplementary Fig. 9). This seems reasonable because KaiA affects not only KaiB–KaiC affinity but also multiple timing processes, such as the phosphorylation state and the ATPase activity of KaiC[40,47]. Based on these observations, we suggest that KaiB–KaiC complex formation is important for the generation and period determination of the damped oscillation in the absence of KaiA. A possible mechanism

would be as follows. Upon light onset, the KaiC–SasA complex formation would initiate and accelerate phosphorylation of RpaA, thereby activating the transcription of *kaiBC*. Subsequent increase in KaiB and KaiC proteins would facilitate KaiB–KaiC complex formation, which in turn reduces the amount of SasA–KaiC complex by substituting SasA with KaiB. The resulting KaiB–KaiC complex binds to CikA, which attenuates RpaA phosphorylation. Thus, the *kaiBC* transcription is reduced to close the TTFL. Although accumulation of KaiB would trigger transition from SasA–KaiC to KaiB–KaiC, it should be noted that the rate of KaiB–KaiC complex formation is slow and relatively insensitive to a change in concentration of the proteins[49]. Instead, the assembly rate of the complex is highly dependent on the biochemical property of KaiC (possibly, ATPase activity in the CI domain)[44], which is altered by the above-mentioned mutations[48].

It is interesting that the *kaiA*-less oscillation and the oscillation in the *kaiC*$^{EE}$ mutant share some similarity in their circadian characteristics: they are less temperature-compensated, subservient to the intact KaiA-including timing system, hypersensitive to photic entraining signals, and more dependent on TTFL[26,50]. They support (i) unnecessity of KaiC phosphorylation for driving imperfect oscillations, and (ii) contribution of KaiC phosphorylation cycle in the intact Kai system for robust circadian timing with KaiA. Nevertheless, there are also striking differences between the two oscillations. The damping rate of the *kaiA*-less oscillation (0.057) is much higher than that of the *kaiC*$^{EE}$ rhythm (0.016). The RpaA-dependent transcriptional rhythm is evident with high amplitude in the *kaiC*$^{EE}$ strain[26], while it is difficult to detect transcriptional rhythms, except for the bioluminescence monitoring system in the *kaiA*$^-$ strain. Even though these results suggest that the *kaiA*-less oscillation is more preliminary and less robust, the period length is more precisely tuned within circadian range than the *kaiC*$^{EE}$ rhythm (~40 h). Apparently, KaiA must be involved in the mechanism to lengthen the period in the *kaiC*$^{EE}$ strain compared with the WT strain. For example, KaiB–KaiC formation rate is decreased by KaiA[44], which might be related to a long period of *kaiC*$^{EE}$. As far as involvement of KaiA is concerned, it is also supported by our observation that introduction of *kaiC*$^{EE}$ mutation less affected the period length in the *kaiA*-less oscillation.

**Mathematical model suggests that the TTFL rhythm can be maintained by resonating with the PTO.** We next constructed a mathematical model for the cyanobacterial clock based on the aforementioned findings to elucidate the role of TTFL. We considered the clock mechanism in the WT strain to be a coupled system consisting of TTFL and PTO (Fig. 6a). We employed the model proposed by Rust et al.[42] for PTO. We simply assumed that the doubly phosphorylated KaiC can bind to KaiB, whereas in the *kaiA*$^-$ mutant, KaiC can form a complex with KaiB. The developed models for the WT and *kaiA*$^-$ strains reproduced stable and damped oscillations, respectively (Fig. 6b). In addition, we assumed that dark environments enhance the degradation rate of *kaiBC* mRNA. The damped rhythm of *kaiA*$^-$ recovered its amplitude by resonating with LD cycles, as observed in Fig. 3. The resonant period of the LD cycle was approximately 24 h, which is the natural period of the *kaiA*$^-$ model (Fig. 6c). Moreover, the *kaiA*$^-$ model reproduced phase responses to dark pulses (Fig. 6d). Note that the response was not a periodical function of the time at dark-stimulation onset because the *kaiA*$^-$ model exhibited damped, but not self-sustained oscillation. At the later timing the dark was exposed, the larger the magnitude of phase shift became. This is because that the oscillation amplitude of the *kaiA*$^-$ model was diminishing.

Moreover, we examined the parameter sensitivity of the rate of phosphorylation $k_{TD}$ and the production rate of the KaiB–KaiC

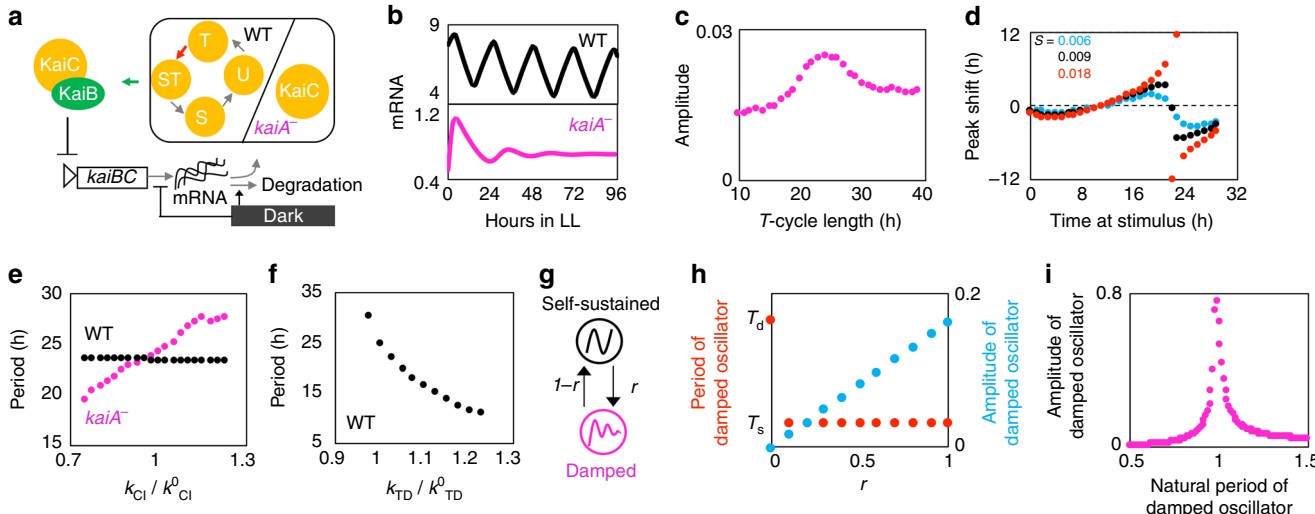

**Fig. 6 Mathematical model for the kaiA-less damped clock system. a** The wild-type (WT) model consists of both the transcription–translation feedback loop (TTFL) and post-translational oscillator (PTO) of Kai, whereas the $kaiA^-$ system features only TTFL. **b** Dynamics of $kaiBC$ mRNA in simulation. **c** Resonance of the $kaiA^-$ model with light–dark (LD) cycles. Amplitude of the mRNA oscillation was measured after exposure to LD cycles with a period of $T$ (h). **d** Phase response of the $kaiA^-$ model to dark pulses with magnitude of $s$. The shifts of the mRNA peak were measured after exposure to a 2-h dark pulse. **e** Dependency of the period length on the rate constant of KaiBC complex formation (corresponding to the green arrow in panel **a**) in both WT and $kaiA^-$ models. **f** Dependency of period length on the rate constant of phosphorylation (corresponding to the red arrow in panel **a**) in the WT model. **g** Simplified system for the coupling of TTFL and PTO. We considered the coupled system of a damped linear oscillator with a natural period of $T_d$ and a phase oscillator with a natural period of $T_s$. The coupling strength was parameterized by $r$ so that the sum of two-way is 1. **h** Dependency of period length and amplitude of the damped oscillator on the coupling strength $r$. **i** Oscillation amplitude of the damped oscillator depends on the natural period of itself in the simplified coupled oscillators. The natural period of the self-sustained oscillator was fixed at 1. The damped oscillator shows higher amplitude when its natural period is closer to 1.

complex $k_{CI}$ to the period of the system. When we altered the production rate of the KaiB–KaiC complex, the oscillation period in $kaiA^-$ was severely affected compared with that of the WT, which is consistent with the experimental findings (Figs. 5f and 6e). Regarding the WT strain, KaiC phosphorylation had a stronger influence on the oscillation period than on the production rate of the KaiB–KaiC complex (Fig. 6f). These results suggest that PTO determines the period in the WT strain, whereas the production rate of the KaiB–KaiC complex plays a larger role in $kaiA^-$ mutants. To confirm the dominancy of PTO, we considered a highly simplified system. We regarded TTFL as a linear damped oscillator and PTO as a phase oscillator, and considered the coupled system of the two oscillators (Fig. 6g). The period of the coupled system was identical to the natural period of a phase oscillator as long as there were interactions among these oscillators, meaning that the self-sustained oscillator perfectly entrained damped rhythms (Fig. 6h). The amplitude of TTFL was determined by how strongly the self-sustained oscillator affected the damped oscillator. The oscillation amplitude also depended on the natural period of TTFL (Fig. 6i), i.e., the agreement of the natural period between these oscillators makes the amplitude of TTFL larger through resonance. This numerical result suggests that the cyanobacterial clock system should take advantage of the TTFL-including damped oscillation with a period of about 24 h to amplify the oscillation amplitude of the whole clock system via resonance.

## Conclusion

In the present study, we observed the damped oscillation of transcriptional output in the absence of KaiA. The output requires the canonical clock components KaiB and KaiC, as well as the master output regulators, namely the SasA/CikA–RpaA

two-component system. The TTFL process is likely involved in generating the damped oscillation, whereas KaiB–KaiC complex formation is, at least in part, important for the period determination process. It is experimentally difficult, however, to confirm that TTFL is at work by monitoring oscillating profiles in KaiBC protein or $kaiBC$ mRNA abundance in $kaiA^-$. In the WT strain, the peak-to-trough ratios of the robust $P_{kaiBC}$ bioluminescence, $kaiBC$ mRNA, and KaiC protein rhythms in LL are ~14, 6, and 2, respectively, under our experimental conditions. In the $kaiA$-null mutant, the peak-to-trough ratio of the bioluminescence is at best 2.3 even for the first peak (Fig. 1). Although linear regression is not evident, if this damping ratio is applied, we would need to detect mRNA and protein cycles with the peak–trough ratio of at best ~ 1.6 and ~1.2, respectively, with much lesser temporal resolution (every 4 h for mRNA/protein vs. every 30 min for bioluminescence) and much higher experimental deviation. Considering this difficulty, however, we examined it in the $kaiA$-null mutant. The profiles of bioluminescence, mRNA, and KaiC in WT are almost the same as previously reported (Supplementary Fig. 10d). The $kaiA^-$ strain showed low-amplitude fluctuating $kaiBC$ and KaiC abundances, as expected, somewhat correlating with the bioluminescence profile (Supplementary Fig. 10e). Thus, these preliminary data could be supportive for the idea of TTFL, while more detailed and multiple experiments are necessary.

Although the damped oscillation was less robust than sustained oscillation and less temperature-compensated, it can maximally resonate to cyclic environments with a period of 1 day. After our finding that TTFL is not essential for circadian rhythms[6], TTFL in the cyanobacterial circadian system has been proposed as a secondary loop to support robustness of the self-sustained post-translational oscillator[37,50,51], forming a coupled positive and negative feedback loops for proper timing[52], and proper

entrainment[50,53]. Our simulation suggests that the TTFL-related damped oscillator can resonate with the post-transcriptional Kai oscillator for robust timing. It also strongly suggests that the KaiB–KaiC-based damped oscillator in the absence of KaiA would be functional, at least as an hourglass timing mechanism. Our study should provide valuable insights into the survival strategies of cyanobacterial species lacking *kaiA* or other bacterial species harboring only *kaiB* and *kaiC* homologs without *kaiA*[11,16,17,54]. The finding also supports an evolutionary hypothesis that a proto-circadian system might evolve without KaiA as a KaiB–KaiC-based damped oscillatory (or hourglass) timing system that can resonate to environmental cycles, and after evolving KaiA, it was organized into the intact KaiABC-based sustained oscillator[55]. The KaiC-binding, KaiB-like domain of SasA is only conserved in cyanobacterial species but not in other bacteria. Thus, transcriptional output mechanism from the KaiBC system would be different in cyanobacteria and other bacteria, such as *Rhodopseudomonas*[19,56]. Since *Prochlorococcus* MED4 is thought to have lost *kaiA* after evolution of the intact KaiABC, its TTFL situation would be more similar to the damped oscillatory system in the *Synechococcus kaiA⁻* mutant strain. In any case, it would be intriguing to address if the resonance effect shown in our study with external cycles of environmental cues with different period length is observed in *Prochlorococcus* MED4 and *Rhodopseudomonas*.

## Methods

**Strains and culture conditions**. We used WT and mutant *Synechococcus* strains, each harboring a bacterial luciferase gene set *luxAB* fused to the D4 promoter[21] as listed and described in Supplementary Table 1. A *kaiA*-inactivated strain with a single stop codon insertion (*kaiA⁻*) was essentially the same as the previously described *kaiA*-inactivated[1,3]. *kaiA⁻* and *kaiB⁻* refer to inactivation of the genes using an insertional stop codon in the ORF to inhibit translation, whereas Δ*kaiA*, Δ*kaiABC*, and Δ*kaiBC* refer to complete deletion of the genes via replacement with an antibiotics-resistance gene. *kaiA*-null strains refer to all of *kaiA⁻* and Δ*kaiA* strains. Details of strains are shown in the following paragraph and in Supplementary Table 1.

ILC976 is an isolated clone from passage culture of the NUC 42 strain[57]. Although most phenotypes featuring damped oscillation when *kaiA* was inactivated are the same, some phenotypes, such as the magnitude of phase shifting, as shown in Fig. 2a, were somewhat different between the past experiment performed in NUC 42 [58] and our present experiments with ILC976. Similarly, ILC128 is essentially identical to NUC43[57], whereas we newly prepared this strain using ILC976 as a host for transformation. pIL764 was constructed to harbor a spectinomycin-resistance gene (Ω cassette) between the segment 1000 bp upstream of the *kaiA* ORF and that 1000 bp downstream of the *kaiC* ORF in a pGEM-T Easy vector (Promega). To construct pIL813 and pIL810, the *XhoI-AflII* site and *EcoRV-BspEI* of the *kaiA*-inactivating plasmid (pCkaiABC derivative renamed pIL1047)[1] were replaced with appropriately point-mutated (nonsense mutations) segments. Note that the single *kaiA⁻* and *kaiB⁻* mutants harbor nonsense mutations (TAA) in the fourth codon (CAA) of *kaiA* or the fifth codon (AAA) of *kaiB*. An additional nonsense mutation (TAG) of *kaiA* in pIL813 was introduced at the 248th codon (AAG). Plasmids for *kaiA;kaiC* double mutants (pIL698, pIL776, pIL777, pIL838, pIL839, and pIL840) were created by introducing point mutations into the *EcoRV-BspEI* segment of the pCkaiABC-based *kaiA*-inactivating plasmid (pIL1047)[1]. To express *kaiBC* constitutively at the moderate level, the promoter-less *kaiBC* segment fused to the promoter region of SynPCC7942_0050 (700-bp upstream region of the gene at −700 to −1: the initiating nucleotide A of the ATG start codon as +1)[37] and an erythromycin-resistance gene derived from pRL271 were inserted into the *Synechococcus* NSIII segment[59]. These segments were cloned into pGEM-T Easy vector (Promega), designated pIL958. Sp^R, Cm^R, Gm^R, and Km^R indicate antibiotics resistance against spectinomycin, chloramphenicol, gentamycin, and kanamycin, respectively.

Cells were cultured on BG-11 solid or liquid medium under LL conditions at 30 °C with fluorescence lamps operated at 30 μmol photon m⁻² s⁻¹. To select and maintain transformants, spectinomycin, chloramphenicol, kanamycin, gentamicin, and/or erythromycin were used.

**Bioluminescence assay**. After cultivation of each strain in BG-11 liquid medium under LL for 2–4 days, 10-μl aliquots of diluted cells (corresponding to an optical density at 730 nm of approximately $1 \times 10^{-5}$) were inoculated onto BG-11 solid medium in 35-mm plates. After 5 days under LL, cells were synchronized to two 12-h:12-h LD cycles, and then bioluminescence was measured under LL at 50 μmol

photon m⁻² s⁻¹ in the presence of 1% decanal solution as a substrate in each plate[60]. The bioluminescence values were measured with a photomultiplier tube H7360-01MOD (Hamamatsu Photonics KK, Japan) and normalized to the number of colonies per plate (counts per colony). Because of the low signal-to-noise ratio in the damped bioluminescence profiles in the *kaiA*-null background, if necessary, we also provided detrended data to show damped oscillation property with the following formula to remove trends with *lag* (=10 h)

$$\text{Bioluminescence}_{\text{Detrended}}\left(t + \frac{\text{lag} - 1}{2}\right)$$
$$= \text{Bioluminescence}(t + \text{lag}) - \text{Bioluminescence}(t). \quad (1)$$

**Phase shifting and resonance experiments**. Cells were prepared and entrained as described above in "Bioluminescence Assay". After entrainment to two LD cycles, the plates were placed in LL, and the measurements of bioluminescence were started. Each plate was removed from the bioluminescence monitoring system transiently and transferred to the dark for 2, 4, or 6 h at hours 4, 8, 12, 16, 20, 24, or 28 in LL. After this dark acclimation, each plate was returned to LL on the bioluminescence monitoring system, and the bioluminescence assay was resumed. The durations of phase shift were calculated by comparing dark-pulse-acclimated bioluminescence rhythm with that without dark pulses. It should be noted that phase information is not available for the above-mentioned detrended data because the moving average method generates a delay. Therefore, we compared peak time of each bioluminescence rhythm (not detrended data but original traces). For resonance experiments, cells were synchronized to two LD cycles, and the bioluminescence was monitored as described above. During the bioluminescence measurement, each plate was transiently removed from the monitoring system to the dark for 2 h four times with an interval of 16, 20, 24, 26, 30, or 32 h to administer dark pulses with different external cycles (T cycle).

**Western blotting**. After cells were cultured to reach OD₇₃₀ of 1–2 in BG-11 liquid media bubbled with air, the strains described in Fig. 4g were inoculated in a flask and grown under batch conditions, while the strains shown in Supplementary Fig. 7 were grown in a continuous cultural system to keep turbidity. In both conditions, cells were entrained to two 12-h:12-h LD cycles and then removed to LL at 30 °C with a light intensity of 50 μmol photon m⁻² s⁻¹. The optical density at OD730 was maintained at 0.25 or 0.4. The bioluminescence from continuous culture (shown in Supplementary Fig. 7a, b) was monitored by a flow system in the presence of 10% decanal as a substrate for luciferase. Cells were harvested at appropriate times and stored at −80 °C after centrifugation. Cell lysate was extracted with sample buffer (25 mM Tris, pH 8.0, 0.5 mM EDTA, and 1 mM DTT)[61]. In total, 5 μg of total proteins were subjected to SDS-PAGE, and Western blotting using anti-KaiA, anti-KaiC antisera[25], and ECL^TM Anti-rabbit IgG Horseradish Peroxide linked the whole antibody (from donkey) NA934V lot#321815 (Amersham Biosciences, UK) at 500-, 5000-, and 5000-fold dilutions, respectively. The chemiluminescence detection was performed using LAS 3000 (Fuji Photo Film CO, Tokyo, Japan), and the image was quantified using ImageJ software (NIH).

**Kai protein complex formation rate in vitro**. Recombinant KaiB and KaiC proteins (final 50 and 200 ng/μl, respectively) were mixed in the presence or absence of KaiA (50 ng/μl), and incubated at 30 °C in reaction buffer (150 mM NaCl, 5 mM MgCl₂, 1 mM ATP, 0.5 mM EDTA, and 50 mM Tris-HCl at pH 8.0). Samples (16 μl) were collected at the indicated times (Fig. 5e, Supplementary Fig. 9) with 4 μl of sample buffer (50% glycerol, 0.05% bromophenol blue, and 0.312 mM Tris-HCl at pH 6.8) for native PAGE[62] and stored at −80 °C. The samples were applied at 10 μl per lane and subjected to native PAGE at 4 °C. After electrophoresis, the gels were stained with Quick CBB (FUJIFILM Wako Pure Chemical Corporation), captured with a scanner, and quantified using ImageJ. Fitting of the kinetic data shown in Fig. 5e was performed via nonlinear least-squares regression with the SciPy library module "curve_fitting" and the following formula to refer to the primary reaction

$$f(t) = \left[\frac{BC}{\text{total}}\right]_{\text{max}} \cdot \left\{1 - \exp(-k \cdot t) \cdot \left(1 - \frac{\left[\frac{BC}{\text{total}}\right]_{\text{initial}}}{\left[\frac{BC}{\text{total}}\right]_{\text{max}}}\right)\right\}. \quad (2)$$

**Model for circadian clock systems in WT and *kaiA*-null strains**. We modeled the circadian machinery in the WT strain as a coupled system of the transcription–translation feedback loop (TTFL) and a post-translational oscillator (PTO). The previously proposed model for an in vitro Kai system was employed for PTO[42]. The values of parameters are identical to the ones in the prior paper. Regarding *kaiA⁻* mutants, PTO was removed from the model for WT, and thus, we considered the three-variable model that forms TTFL. The sets of equations and parameters are shown in the Supplemental text. The model was numerically solved via the Runge–Kutta fourth-order method with a time interval of 0.01.

**Simplified model**. We simply modeled the coupling of TTFL and PTO as a coupled system of a phase oscillator and a linear damped oscillator as follows

$$\ddot{x} + 2\lambda\dot{x} + \omega_d^2 x = I\cos\theta, \tag{3}$$

$$\dot{\theta} = \omega_s + \varepsilon\sin\theta \cdot x. \tag{4}$$

We set $\lambda = 0.1$, $\omega_d = \frac{2\pi}{1.2}$ and $\omega_s = 2\pi$, meaning that the natural periods of a damped oscillator $T_d$ and a self-sustained oscillator $T_s$ are 1.2 and 1, respectively. The model was numerically solved via the Runge–Kutta fourth-order method with a time interval of 0.01.

**Equations and parameters of the model for circadian clock in WT**. We added TTFL into the PTO model proposed by Rust et al.[42] as follows:

$$\dot{M} = \frac{k}{K_M + I^n} - d_M \cdot M \tag{5}$$

$$\dot{U} = kTU \cdot T + kSU \cdot S - kUT \cdot U - kUS \cdot U - V \cdot \frac{U}{K + U} - Vd \cdot U + Ks \cdot M \tag{6}$$

$$\dot{T} = kUT \cdot U + kDT \cdot D - kTU \cdot T - kTD \cdot T - V \cdot \frac{T}{K + T} - Vd \cdot T \tag{7}$$

$$\dot{D} = kTD \cdot T + kSD \cdot S - kDT \cdot D - kDS \cdot D - V \cdot \frac{D}{K + D} - Vd \cdot D \tag{8}$$

$$\dot{S} = kUS \cdot U + kDS \cdot D - kSU \cdot S - kSD \cdot S - V \cdot \frac{S}{K + S} - Vd \cdot S \tag{9}$$

$$\dot{I} = k_I \cdot D - d_I \cdot I \tag{10}$$

$$kXY = + \frac{k_{XY}^A A(S)}{K_{\frac{1}{2}} + A(S)} \tag{11}$$

$$A(S) = \max[0, [KaiA] - 2S], \tag{12}$$

where $U$, $T$, $S$, and $D$ are the amounts of unphosphorylated, T432-phosphorylated, S431-phosphorylated, and doubly phosphorylated KaiC, respectively. Rust et al. proposed the rate constants involved in KaiC phosphorylation cycles based on in vitro experiments. We used the same values of parameters as follows: $k_{UT}^0 = k_{TD}^0 = k_{SD}^0 = k_{US}^0 = 0\,h^{-1}$, $k_{TU}^0 = 0.21\,h^{-1}$, $k_{DT}^0 = 0\,h^{-1}$, $k_{DS}^0 = 0.31\,h^{-1}$, $k_{SU}^0 = 0.11\,h^{-1}$, $k_{UT}^A = 0.479077$, $k_{TD}^A = 0.212923\,h^{-1}$, $k_{SD}^A = 0.505692\,h^{-1}$, $k_{US}^A = 0.0532308\,h^{-1}$, $k_{TU}^A = 0.0798462$, $k_{DT}^A = 0.173\,h^{-1}$, $k_{DS}^A = -0.31385$, $k_{SU}^A = -0.133077\,h^{-1}$, $[KaiA] = 1.3\,\mu M$, $K_{1/2} = 0.43\,\mu M$, $V = 0.0078\,\mu M\,h^{-1}$, $K = 0.1\,\mu M$. We chose the rest of the value of parameters so that the oscillation period of this system is approximately 24 h: $n = 10$, $k = 0.5\,h^{-1}$, $K_M = 1\,\mu M$, $d_M = 0.0385\,h^{-1}$, $k_I = 0.5\,h^{-1}$, $d_I = 0.5\,h^{-1}$.

When we considered dark conditions, the evolutional equation of $M$ was replaced with

$$\dot{M} = \frac{k}{K_M + I^n} - (d_M + s) \cdot M. \tag{13}$$

The parameter $s \geq 0$ indicates the enhancement of degradation of the *kaiBC* transcript. $s(t)$ is a square wave under LD cycles.

Note that this model did not exhibit stable limit cycle oscillations when $t$ was sufficiently large. This is possible because we oversimplified the genetic regulations. For example, our model does not consider other pathways mediated by SasA and LabA[35]. Nevertheless, we believe that the significant dependency of the phosphorylation rate on the period generally holds, regardless of the details of the regulatory network.

**Equations and parameters of the model for circadian clock in *kaiA*⁻**. We simplified the WT model to develop the model for the *kaiA*⁻ strain. The dynamics of PTO were merely replaced with the synthesis and degradation of KaiC in this model. The reduced model is

$$\dot{M} = \frac{k}{K_M + I^n} - d_M \cdot M \tag{14}$$

$$\dot{C} = k_c \cdot M - d_C \cdot C \tag{15}$$

$$\dot{I} = k_I \cdot C - d_I \cdot I. \tag{16}$$

We set $k_c = 0.5\,h^{-1}$ and $d_C = 0.25\,h^{-1}$. The values of the other parameters were identical to those in the WT model. This model is equivalent to that proposed by Griffith[63]. Kurosawa et al.[64] found that this model can display both self-sustained and damping oscillations depending on the values of parameters.

**Damped oscillator modeling of bioluminescence signals**. To quantify the experimental data, damped oscillator modeling was carried out for the bioluminescence signals. As a generic model for linear damped oscillator with additive white noise, the following stochastic differential equations are introduced[22]

$$\dot{x} = -\lambda \cdot x - \omega \cdot y + \xi_x, \tag{17}$$

$$\dot{y} = \omega \cdot x - \lambda \cdot y + \xi_y, \tag{18}$$

where $\xi_x$ and $\xi_y$ are independent Gaussian noises satisfying $\langle\xi_x\rangle = 0$, $\langle\xi_y\rangle = 0$, $\langle\xi_x(t+\tau)\cdot\xi_x(t)\rangle = 2D\delta(\tau)$, $\langle\xi_y(t+\tau)\cdot\xi_y(t)\rangle = 2D\delta(\tau)$, $\langle\xi_x(t+\tau)\cdot\xi_x(t)\rangle = 0$ ($\langle\cdot\rangle$ denotes time average, $\delta(\tau)$ is the Dirac's delta function, and $D$ represents the noise intensity. Without noise, the system gives rise to damped oscillations with damping rate $\lambda$ and angular frequency $\omega$. With noise, the system is continuously perturbed and exhibits a noisy periodic behavior. The present system provides a standard approach to model a stochastic gene expression[65]. Its autocorrelation function, defined as $C_M(\tau) = \langle x(t+\tau)\cdot x(t)\rangle = \langle y(t+\tau)\cdot y(t)\rangle$, is derived by Westermark et al.[22]

$$C_M(\tau) = \frac{D}{\lambda}e^{-\lambda\tau}\cos\omega\tau. \tag{19}$$

The stochastic linear damped oscillator model has three unknown parameters $\{\omega, \lambda, D\}$. Our experimental data were fitted to the damped oscillator model by optimizing the three unknown parameters as follows. First, the detrended bioluminescence signal was normalized in such a way that the signal has zero mean and unit variance. Second, with respect to the normalized bioluminescence signal $\{z_t: t = 1,2,..,M\}$, the autocorrelation function $C_B(k)$ (with the time lag of $k$ sampling intervals) was computed as

$$C_B(k) = \frac{1}{M-k}\sum_{t=1}^{M-k}(z_t - \bar{z})(z_{t+k} - \bar{z}), \tag{20}$$

where $\bar{z} = \frac{1}{M}\sum_{t=1}^{M}z_t$ represents the mean value. The autocorrelation function detects periodicity in the bioluminescence signal, where the time lag that points to the first peak roughly corresponds to the period length of the signal. Third, the three parameters $\{\omega, \lambda, D\}$ of the damped oscillator model were optimized so that its autocorrelation function $C_M(\tau)$ is fitted to that of the bioluminescence signal $C_B(k)$. To deal with the experimental autocorrelation function $C_M(\tau)$, whose oscillation center is sometimes deviated from zero, the model function is modified as $C_M(\tau) = C_0 + (D/\lambda)e^{-\lambda\tau}\cos\omega\tau$ and the four parameters $\{\omega, \lambda, D, C_0\}$ were optimized. The time lag of up to $\tau < 40\,h$ was utilized, since estimates of the autocorrelations for larger time lags become poor ($\tau < 50\,h$ for long-period data). For the optimization, we used *lsqcurvefit* subroutine of the MATLAB Statistical Toolbox (Mathworks, R2019a). Initial guesses were used as described by Westermark et al.[22]. Namely, the estimated autocorrelation $C_B(k)$ was smoothed by a low-pass filter, logarithms of the absolute values were taken, and the peaks were extracted by a standard peak-picking algorithm. Initial guess for the damping rate was obtained by least-squares fitting of the logarithms of the peak magnitudes. The averaged peak interval was used as the initial guess for the angular frequency. To assure accuracy of the fitting, optimized parameters, which give the correlation coefficient of the autocorrelation functions between the model and the experiment greater than 0.9, were considered reliable.

For the bioluminescence signals of the WT, *kaiA*⁻, and double mutants, the damped oscillator modeling was carried out. In Supplementary Fig. 3, autocorrelation functions of the experimental data $C_B(k)$ (black) and those of the fitted model $C_M(\tau)$ (red) are compared. The model captured the basic feature of the experimental curve fairly well. Table S2 summarizes the estimated parameters of the intrinsic period $\tau_{\text{int}} = 2\pi/\omega$ and the damping rate $\lambda$.

To examine accuracy of the period estimates, the present model fitting was applied to artificial data sets generated from the linear damped oscillator model described above. Since the oscillation periods are known a priori, the estimation errors are given by deviations of the estimated periods from the true ones. In addition to the modeling fitting approach, the autocorrelation function analysis and the peak-to-peak interval were also tested as alternative methods to estimate the periods. To simulate the artificial data, the damping rate was increased from $\lambda = 0.01$ to 0.2. The periods $\tau_{\text{int}}$ were uniformly distributed between 23.5 and 24.5 h, and the noise level was set to $D = 0.0001$. In each signal, 500 time points were sampled with an interval of 20 min. For each damping rate, the errors were averaged over 100 data sets. The results are shown in Supplementary Fig. 4. As the damping rate was increased, the estimation error increased monotonously for all three methods. This is because the period information is lost quickly as the damping effect is strengthened.

**Materials availability**. All materials are available in the paper or upon request.

**Reporting summary**. Further information on research design is available in the Nature Research Reporting Summary linked to this article.

## Data availability

The source data underlying Figs. 1–5 and Supplementary Figs. 1–9 are provided as a Source Data file. Codes of two mathematical models in Fig. 6 are available at https://github.com/hito1979/NatCommun2020.

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

## Acknowledgements

We thank K. Miwa and T. Kondo for providing recombinant mutant KaiC proteins, C.H. Johnson, and Y. Xu for valuable discussion. We also thank Y. Murayama, K. Kawasaki, and other members of the Iwasaki Laboratory for valuable comments and technical support. This study was supported in part by a Grant-in-Aid for Scientific Research (KAKENHI) from JSPS (Grant nos. 18K19349 and 23657138 to H. Iwasaki, 18H05474 to H. Ito, and 17H06313 to I.T.T.). The authors would like to thank Enago for the English language review.

## Author contributions

N.K. designed the research, performed experiments, constructed the mathematical models, analyzed the data, and wrote the paper. H. Iwasaki contributed to design the research, provided materials and equipment for this study, analyzed the data, and wrote the paper. H. Ito constructed the mathematical models, analyzed the data, and wrote the paper. I.T.T. quantified bioluminescence data by the model-fitting procedures, analyzed the data, and wrote the paper.

## Competing interests

The authors declare no competing interests.
