## [Peer Review File · Nature Communications]

Reviewers' Comments:

Reviewer #1:

Remarks to the Author:

This is an interesting and important study that reveals an underlying damped oscillator with a near-24 h timescale based on transcriptional feedback that becomes apparent when the *kaiA* gene is disrupted in *S. elongatus*. This has evolutionary relevance because the *kaiB* and *kaiC* genes predate the emergence of *kaiA*, and this gives a perspective the fundamental question of how clocks evolved.

This is a carefully done study with high data quality. My concerns / suggestions do not involve new experiments, but should be addressed in the text because they impact the interpretation of the results.

1. There are a lot of similarities here with the Qin et al. 2010 study analyzing the properties of the TTFL-based rhythm in the KaiC-EE mutant, showing that it is not temperature compensated and is subservient to the post-translational timing system in the wildtype. This study should be discussed and contrasted with the results here, since it appears that the *kaiA*- mutation is essentially revealing another aspect of the TTFL-only system.

2. The authors write "However, because of difficulties regarding cell culturing and genetic treatment in *Prochlorococcus*, direct evidence of a non-self-sustaining timer is not yet experimentally demonstrated." Direct evidence of a non-self-sustaining rhythm in KaiC phosphorylation in *Prochlorococcus* was published in Chew et al 2018

3. I do not find the experimental evidence involving the impact of *sasA* and *cikA* mutations on the *kaiA*-less oscillation to be particularly convincing. These mutations also have profound impacts on the amplitude of the intact *kaiABC* oscillator, so I don't see how this can be taken as support for the transcriptional basis of the dampened oscillation. The experimental manipulation of the *kaiBC* promoter is much more compelling.

4. The authors write "The result that the KaiCAA mutant protein cannot be substituted with the hypophosphorylated KaiC without KaiA is also interesting. Although the majority of the KaiC protein is dephosphorylated, we suggest that some KaiC molecules undergo KaiA-independent auto-phosphorylation, making them important for driving the damped oscillation, whereas it was difficult under our experimental conditions to detect residual phosphorylated KaiC via Western blotting". An alternative explanation (that I find more likely) is that KaiC-AA may not be a faithful mimic of the properties of unphosphorylated KaiC. For example it is known that KaiC-AA has abnormally high ATPase activity.

5. The intriguing observation that the period in KaiC-EE -*kaiA* is much shorter than in KaiC-EE +*kaiA* may be explicable in terms of the authors' model that the kinetics of BC complex formation is a key timing process for this oscillation. For example, Lin et al. 2014 showed that the presence of KaiA reduced the ability of KaiC-EE to bind to KaiB (can be compared with Fig 5E here)

7. The modeling result is nice and makes intuitive sense, however I found the statement "This numerical result suggests that the damped oscillation of TTFL with a period of approximately 24 h is the key mechanism of the definite circadian rhythm with a large amplitude" troubling. Strains that lack transcriptional feedback (*P_{trc}::kaiBC* or other constitutive promoter) have high amplitude output rhythms. This seems to contradict this claim. What happens in the model if the authors simulate a constitutively expressed KaiC? Is the amplitude of output rhythms impacted?

Reviewer #2:

Remarks to the Author:

Dear Authors, dear Editor,

The work by Kawamoto et al. suggests the novel finding of KaiA-less oscillations in the model cyanobacterium *Synechococcus elongatus*. Cyanobacteria are special among prokaryotes possessing a true circadian clock which is made of three proteins, KaiA, KaiB and KaiC. This Kai protein clock is very unique, however, it enables important findings on general principles of circadian systems in any kingdom of life. So far, it has been believed (and published many times) that the absence of any of the three genes including kaiA abolishes oscillations. Thus, the experimentally documented damped oscillations now identified by Kawamoto are exciting and open a new view on the circadian clock mechanism in cyanobacteria. The observed damped oscillations nicely resonate with external circadian cycles, Zeitgeber. Interestingly, a phosphorylation cycle of KaiC is not necessary. However, mutating kaiB, kaiC and sasA each abolished damped oscillations. The experimental findings of Kawamoto et al. are well supported by a mathematical model simulating a damped oscillator in cyanobacteria based on KaiB and KaiC only. Personally, I highly appreciate this damped oscillator concept for the cyanobacterial circadian systems. However, in order to proof this so far hypothesis only, several points have to be addressed. My major claims are:

1) a better experimental / statistical verification of the very very weak damped oscillations. More intensive data analysis is needed, for example data fitting using a model of a generic linear damped oscillator with additive white noise, e.g. Westermarck et al. 2009, because "Noise can generate well-defined oscillations in a damped oscillator.". Bioluminescence reporters for P-kaiBC have been observed/studied before which led to an opposite conclusion in the past. This contrast has to be addressed more deeply, and better discussed. Following this line, in Figure S2: What is meant by "Because the second peaks were not observed reproducibly, period lengths were not calculated." Were the second peaks frequently not observed reproducibly? There is no mention of this in the main text and it would be appropriate to mention it.

2) further experimental proof demonstrating the TTFL mechanism, e.g. RT-PCR for kaiBC mRNAs. Obviously, without a functional KaiA the protein oscillator is not able to perform circadian phosphorylation rhythms of KaiC. However, there is no further evidence that instead a TTFL is responsible for the observed damped oscillations. Either the kaiBC transcript level or KaiB and KaiC protein levels are expected to oscillate in the KaiA-absent scenario which should be shown experimentally.

3) more extensive discussion including further references, e.g. minimal Kai-based systems e.g. by Schmelling et al. 2017; the concept of a damped oscillator adopted from e.g. the human circadian clock; TTFL models for *S. elongatus* e.g. by Hertel et al. 2010 and Zwicker et al. 2010. Further, the paragraph on KaiBC complex formation ends with the statement that this complex is important for generation and period determination of the damped oscillation in the absence of KaiA. In my opinion, the explanation for this is a little too brief. The observation is first and foremost rather obvious and therefore it should probably be supported by further discussion or even further experiments.

4) reproducible (and ideally open access e.g. protocols.io, and code for modeling at github or zenodo) description of experimental methods, e.g. bioluminescence measurements. Bioluminescence Assay is insufficiently described. What is the decanal solution for, at what wavelength was the OD measured, were the cells excluded that did not show a signal? It is not clear what the purpose of trend elimination is. The original publication of the bioluminescence assay does not describe comprehensively what and how luminescence is evaluated and measured which is key for the major findings of the manuscript. Further, the experiment on the entrained circadian clock is also not described in detail. Are the light pulses randomly selected after 2, 4 and 6 hours or are there additional reasons for this? The conclusion regarding higher sensitivity to dark stimuli is not described sufficiently.

Further there are several minor concerns which should be considered as well:

- kaiA-, kaiA-less, and, kaiA- and ∂ kaiA should be used consistently throughout the manuscript,

and might be explained more in detail

- lines 89 - 94: very difficult to read and to understand

- lines 95 - 97: could be discussed more deeply

- lines 166 - 167: please explain why more KaiBC interferes with the bioluminescence? Might be also important for data analysis.

- lines 191 - 192: I do not understand this sentence, please explain further.

General

Based on reviewer's suggestions, we largely modified the text.

Main figures 1-6 are essentially same as previously submitted.

Supplementary Materials

Supplementary Method: We added newly performed fitting analysis ("Damped oscillator modeling of bioluminescence signals" section). For this analysis, we added a new co-author, Dr. Isao T. Tokuda, who is an expert of this field.

Table S1: References and the legend, including detailed strain construction info, were corrected.

Table S2: Newly added analysed dataset to summarize calculated periods and damping rates of each bioluminescence profile by three different methods for comparison.

Figure S1: Newly added experimental data to show bioluminescence profiles at low light intensity.

Figure S3: Newly added auto-correlation data to validate plausibility of the fitting method.

Figure S4: Newly added analysed data (dependence of the estimation error of period τ_{int} on the damping rate).

Figures S2, S5, S6, S7, S8 and S9 are same as previously submitted Figures S1, 2, 3, 4, 5 and 6, respectively.

Response to Reviewer #1

1. There are a lot of similarities here with the Qin et al. 2010 study analyzing the properties of the TTFL-based rhythm in the KaiC-EE mutant, showing that it is not temperature compensated and is subservient to the post-translational timing system in the wildtype. This study should be discussed and contrasted with the results here, since it appears that the kaiA-mutation is essentially revealing another aspect of the TTFL-only system.

We thank the reviewer for the insightful comment. We agree that the both systems (*kaiA*-null and *kaiC-EE*) are similar, such as (1) primarily based on the TTFL process, (2) less temperature compensated property, and (3) higher sensitivity to a photic (dark) entraining signal to reset the clock. On the other hand, damping rate is much higher (more damped) in the *kaiA*-null strain. Thus, we can only detect rhythmicity with high-resolution bioluminescence reporter assays, while in *kaiC-EE* mutant we can detect robust transcriptional rhythms at mRNA levels (northern and microarrays). In addition, we do not use the "TTFL-only" to describe the EE mutant, since we cannot exclude a possibility that some residual posttranslational oscillatory functions, such as subunit interactions and ATPase activity, remain in the mutant clock system. Thus, we added the text as follows.

"It is interesting that the *kaiA*-less oscillation and the oscillation in the *kaiC^{EE}* mutant share some similarity in their circadian characteristics: they are less temperature compensated, subservient to the intact KaiA-including timing system, hypersensitive to photic entraining signals, and more dependent on TTFL^{26,50}. They support (i) unnecessary of KaiC phosphorylation for driving imperfect oscillations, and (ii) contribution of KaiC phosphorylation cycle in the intact Kai system for robust circadian timing with KaiA. Nevertheless, there are also striking difference between the two oscillations. Damping rate of the *kaiA*-less oscillation (0.057) is much higher than that of the *kaiC^{EE}* rhythm (0.016). The RpaA-dependent transcriptional rhythm is evident with high amplitude in the *kaiC^{EE}* strain²⁶, while it is difficult to detect transcriptional rhythms except for the bioluminescence monitoring system in the *kaiA*⁻ strain. Even though these results suggest that the *kaiA*-less oscillation is more preliminary and less robust, the period length is more precisely tuned within circadian range than the *kaiC^{EE}* rhythm (~40 h). Apparently, KaiA must be involved in the mechanism to lengthen the period in the *kaiC^{EE}* strain compared with the wild type strain. For example, KaiB-KaiC formation rate is decreased by KaiA⁴⁴, which might be related to long period of *kaiC^{EE}*. As far as involvement of KaiA, it is also supported by our observation that introduction of *kaiC^{EE}* mutation less affected the period length in the *kaiA*-less oscillation." (lines 286-298)

2. The authors write "However, because of difficulties regarding cell culturing and genetic treatment in *Prochlorococcus*, direct evidence of a non-self-sustaining timer is not yet experimentally demonstrated." Direct evidence of a non-self-sustaining rhythm in KaiC phosphorylation in *Prochlorococcus* was published in Chew et al 2018.

We appreciate the reviewer's suggestion. Considering this and the other reviewer's comments, we included some additional references on both *Prochlorococcus* and *Rhodospseudomonas* and modified the text as follows.

"Differing from KaiC in *Synechococcus*, KaiA is not essential for enhancing the basal auto-phosphorylation activity of the KaiC homologue in *Prochlorococcus* MED4 and *Rhodospseudomonas*^{12,16}. In both species KaiC homologues undergoes phosphorylation in the light and dephosphorylation in the dark^{16,17}. Based on these results, prior research

discussed the possibility of a non-self-sustaining timing system in cyanobacterial and purple bacterial species lacking *kaiA*^{11, 12, 16, 17, 18}. More recently, a comparative bioinformatics study by Schmelling *et al*¹⁹ reported a possible conserved gene network composed of *kaiB*, *kaiC*, *sasA*, *rpaA*, *rpaB*, *ldpA*, *cpmA* and *ircA* among cyanobacterial species, and proposed these genes as possible components of the prototypic hourglass-like timing system. There are a couple of possible mechanisms for the timing system other than a self-sustained oscillator from a mathematical viewpoint. One mechanism is the hourglass model, which can respond to periodical environments but does not exhibit any oscillations under constant conditions. The other possibility is damped oscillation, which can display oscillations under constant conditions, although its amplitude can decay exponentially. In both *Prochlorococcus* and *Rhodospseudomonas*, these possibilities have not been experimentally validated.” (lines 42-54)

3. *I do not find the experimental evidence involving the impact of sasA and cikA mutations on the kaiA-less oscillation to be particularly convincing. These mutations also have profound impacts on the amplitude of the intact kaiABC oscillator, so I don't see how this can be taken as support for the transcriptional basis of the dampened oscillation. The experimental manipulation of the kaiBC promoter is much more compelling.*

We understand the point. However, if this logic is more generally applied, we cannot discuss *kaiA;kaiC* mutant strain, neither, since nullification of *kaiC* already nullifies the oscillation. Instead, we think it important and informative to confirm that additional *kaiA*-inactivation in *sasA* or *cikA* background further dampened the oscillation. Thus, we weakened discussion about CikA as follows.

“KaiB, KaiC, and the SasA-RpaA two-component system are likely essential for driving *kaiA*-less damped oscillation, and CikA might be also involved in its modulation.” (lines 189-190)

4. *The authors write "The result that the KaiCAA mutant protein cannot be substituted with the hypophosphorylated KaiC without KaiA is also interesting. Although the majority of the KaiC protein is dephosphorylated, we suggest that some KaiC molecules undergo KaiA-independent auto-phosphorylation, making them important for driving the damped oscillation, whereas it was difficult under our experimental conditions to detect residual phosphorylated KaiC via Western blotting". An alternative explanation (that I find more likely) is that KaiC-AA may not be a faithful mimic of the properties of unphosphorylated KaiC. For example it is known that KaiC-AA has abnormally high ATPase activity.*

We thank the reviewer’s comment. We added the following text, according to the comment, citing Mutoh *et al.* 2013.

“In addition, because of the higher ATPase activity of KaiC^{AA} than intact KaiC and KaiC^{EE}, KaiC^{AA} almost does not bind to KaiB *in vitro*⁴⁰. Alternatively, we cannot eliminate a possibility that the KaiC^{AA} mutation is not a simple mimic of unphosphorylated form of KaiC, and may have additional abnormalities in its biochemical property: its abnormally higher ATPase may manifest this side effect.” (lines 245-249)

5. *The intriguing observation that the period in KaiC-EE -kaiA is much shorter than in KaiC-EE +kaiA may be explicable in terms of the authors' model that the kinetics of BC complex formation is a key timing process for this oscillation. For example, Lin et al. 2014 showed that the presence of KaiA reduced the ability of KaiC-EE to bind to KaiB (can be compared with Fig 5E here)*

We thank the reviewer’s thoughtful comment. As described above (to the comment 1), we modified the text as follows:

“Even though these results suggest that the *kaiA*-less oscillation is more preliminary and less robust, the period length is more precisely tuned within circadian range than the *kaiC^{EE}* rhythm (~40 h). Apparently, KaiA must be involved in the mechanism to lengthen the period in the *kaiC^{EE}* strain compared with the wild type strain. For example, KaiB-KaiC formation rate is decreased by KaiA⁴⁴, which might be related to long period of *kaiC^{EE}*. As far as involvement of KaiA, it is also supported by our observation that introduction of *kaiC^{EE}* mutation less affected the period length in the *kaiA*-less oscillation.” (lines 293-298)

6. *The modeling result is nice and makes intuitive sense, however I found the statement "This numerical result suggests that the damped oscillation of TTFL with a period of approximately 24 h is the key mechanism of the definite circadian rhythm with a large amplitude" troubling. Strains that lack transcriptional feedback (Ptrc::kaiBC or other constitutive promoter) have high amplitude output rhythms. This seems to contradict this claim. What happens in the model if the authors simulate a constitutively expressed KaiC? Is the amplitude of output rhythms impacted?*

We appreciate this reviewer’s comment. As suggested, our model also showed PTO oscillations under constitutive promoter without showing any rhythms at the *kaiBC* mRNA levels. (Figure presented below)

Thus, we agree to the opinion that our previous statement exaggerated the importance of resonance. Nevertheless, the numerical simulation suggests that the oscillatory feature of TTFL network should contribute to enhancement of amplitude of the whole circadian system. To avoid this confusion, we revised the statement as followings,

“This numerical result suggests that the cyanobacterial clock system should take advantage of the TTFL-including damped oscillation with a period of about 24 h to amplify the oscillation amplitude of the whole clock system via resonance.” (lines 325-327)

Response to Reviewer #2

1-1) A better experimental / statistical verification of the very very weak damped oscillations. More intensive data analysis is needed, for example data fitting using a model of a generic linear damped oscillator with additive white noise, e.g. Westermark *et al.* 2009, because “Noise can generate well-defined oscillations in a damped oscillator”.

We agree to the reviewer’s suggestion, and thus invited Dr. Isao T. Tokuda as a new co-author, who is an expert in this field and has collaborated with Dr. Westermark. Then, we totally re-evaluated the bioluminescence profiles with the Westermark’s method and compared with our previous dataset. We described detailed info on newly added Supplementary Information (text and newly added Supplementary Figures 3 and 4, and Supplementary Table 2) as follows:

“Damped oscillator modeling of bioluminescence signals:

To quantify the experimental data, damped oscillator modeling was carried out for the bioluminescence signals. As a generic model for linear damped oscillator with additive white noise, the following stochastic differential equations are introduced⁵:

$$\begin{aligned}\dot{x} &= -\lambda \cdot x - \omega \cdot y + \xi_x, \\ \dot{y} &= \omega \cdot x - \lambda \cdot y + \xi_y,\end{aligned}$$

where ξ_x and ξ_y are independent Gaussian noises satisfying $\langle \xi_x \rangle = 0$, $\langle \xi_y \rangle = 0$, $\langle \xi_x(t + \tau) \cdot \xi_x(t) \rangle = 2D\delta(\tau)$, $\langle \xi_y(t + \tau) \cdot \xi_y(t) \rangle = 2D\delta(\tau)$, $\langle \xi_x(t + \tau) \cdot \xi_x(t) \rangle = 0$ ($\langle \cdot \rangle$ denotes time average, $\delta(\tau)$ is the Dirac's delta function, and D represents the noise intensity). Without noise, the system gives rise to damped oscillations with damping rate λ and angular frequency ω . With noise, the system is continuously perturbed and exhibits a noisy periodic behavior. The present system provides a standard approach to model a stochastic gene expression⁶. Its autocorrelation function, defined as $C_M(\tau) = \langle x(t + \tau) \cdot x(t) \rangle = \langle y(t + \tau) \cdot y(t) \rangle$, is derived as Westermark *et al.*⁵

$$C_M(\tau) = \frac{D}{\lambda} e^{-\lambda\tau} \cos \omega\tau.$$

The stochastic linear damped oscillator model has three unknown parameters $\{\omega, \lambda, D\}$. Our experimental data were fitted to the damped oscillator model by optimizing the three unknown parameters as follows. First, the detrended bioluminescence signal was normalized in such a way that the signal has zero mean and unit variance. Second, with respect to the normalized bioluminescence signal $\{z_t : t=1,2,\dots,M\}$, the autocorrelation function $C_B(k)$ (with the time lag of k sampling intervals) was computed as

$$C_B(k) = \frac{1}{M-k} \sum_{t=1}^{M-k} (z_t - \bar{z})(z_{t+k} - \bar{z}),$$

where $\bar{z} = \frac{1}{M} \sum_{t=1}^M z_t$ represents the mean value. The autocorrelation function detects periodicity in the bioluminescence signal, where the time lag that points to the first peak roughly corresponds to the period length of the signal. Third, the three parameters $\{\omega, \lambda, D\}$ of the damped oscillator model were optimized so that its autocorrelation function $C_M(\tau)$ is fitted to that of the bioluminescence signal $C_B(k)$. To deal with the experimental autocorrelation function $C_M(\tau)$, whose oscillation center is sometimes deviated from zero, the model function is modified as $C_M(\tau) = C_0 + (D/\lambda)e^{-\lambda\tau} \cos \omega\tau$ and the four parameters $\{\omega, \lambda, D, C_0\}$ were optimized. The time lag of up to $\tau < 40$ h was utilized, since estimates of the autocorrelations for larger time lags become poor ($\tau < 50$ h for

long period data). For the optimization, we used *lsqcurvefit* subroutine of the MATLAB Statistical Toolbox (Mathworks, R2019a). Initial guesses were used as described by Westermarck *et al.*⁵ Namely, the estimated autocorrelation $C_B(k)$ was smoothed by a low-pass filter, logarithms of the absolute values were taken, and the peaks were extracted by a standard peak-picking algorithm. Initial guess for the damping rate was obtained by least-squares fitting of the logarithms of the peak magnitudes. The averaged peak interval was used as the initial guess for the angular frequency. To assure accuracy of the fitting, optimized parameters, which give correlation coefficient of the autocorrelation functions between the model and the experiment larger than 0.9, were considered reliable.

For the bioluminescence signals of the wild-type, $kaiA^-$, and double mutants, the damped oscillator modeling was carried out. In Supplementary Fig. 3, autocorrelation functions of the experimental data $C_B(k)$ (black) and those of the fitted model $C_M(\tau)$ (red) are compared. The model captured basic feature of the experimental curve fairly well. Table S2 summarizes the estimated parameters of the intrinsic period $\tau_{int}=2\pi/\omega$ and the damping rate λ .

To examine accuracy of the period estimates, the present model fitting was applied to artificial data sets generated from the linear damped oscillator model described above. Since the oscillation periods are known *a priori*, the estimation errors are given by deviations of the estimated periods from the true ones. In addition to the modeling fitting approach, the autocorrelation function analysis and the peak-to-peak interval were also tested as alternative methods to estimate the periods. To simulate the artificial data, the damping rate was increased from $\lambda=0.01$ to 0.2. The periods τ_{int} were uniformly distributed between 23.5 h and 24.5 h and the noise level was set to $D=0.0001$. In each signal, 500 time points were sampled with an interval of 20 min. For each damping rate, the errors were averaged over 100 data sets. The results are shown in Supplementary Fig. 4. As the damping rate was increased, the estimation error increased monotonously for all three methods. This is because the period information is lost quickly as the damping effect is strengthened.” (Supplementary Method)

Based on this analysis, we largely modified the main text at the section of period-determination process as follows:

“We compared three methods for estimating periods from bioluminescence signals: (i) damped oscillator model fitting by the method of Westermarck *et al.*²² (Supplementary Fig. 3), (ii) autocorrelation function analysis, and (iii) peak-to-peak interval. For the WT data, the averaged period was estimated to be 25.1 hour by all three methods. The averaged period of $kaiA^-$, on the other hand, was estimated to be 25.8 h, 24.8 h, and 24.0 h by the model fitting, the autocorrelation, and the first peak-to-peak interval, respectively (Supplementary table 2). To verify precision of the period estimates, the three methods were applied to artificial data sets generated from linear damped oscillators, the periods of which were known *a priori*. As described in detail in supplementary information, the estimation error is given by deviation of the estimated periods from the true ones. As the damping rate was increased, the estimation error increased monotonously for all three methods. This is because the period information is lost quickly by a strong damping. In the model fitting, the estimation error increased to 1 h at a damping rate of $\lambda=0.05$ and reached to 2 h at $\lambda=0.1$ (Supplementary Fig. 4). The autocorrelation analysis gave even larger errors. The peak-to-peak interval, on the other hand, produced results comparable to those of the model fitting, when initial two peak-to-peak intervals were averaged as the period estimate. When only the first peak-to-peak interval was used, the estimation errors became much smaller than the model fitting, especially for a damping rate between 0.025 and 0.15 (Supplementary Fig. 4). In noisy damped data, signals are attenuated quickly, while the noise effect becomes non-negligible. This lowers the signal-to-noise ratio over time. For this reason, the first peak-to-peak interval provides the most reliable period information than the other two methods that average long-term properties of the attenuated signals. For a precise estimation of period from damped oscillators, it is more advantageous to utilize the portion, in which the signal is least attenuated. According to our estimate, the damping rate of $kaiA^-$ was approximately 0.05, *i.e.*, within the range where periods are most precisely estimated by the first peak-to-peak interval. Therefore, in the following period analysis, the time intervals between the first and second peaks are regarded as the period.” (lines 122-143, main text)

Furthermore, based on our new calculation, we modified the values of damping rate of each strain, such as:

“The damping rates were 0.083, 0.069, and 0.091 for S157P, F470Y, and EE mutations in $kaiA^-$ background, respectively, and the amplitude was quickly reduced to about 10-15% per each period. On the other hand, interestingly, the damping rate of $kaiA^-;kaiC^{R321Q}$ was lower (0.027), and the amplitude remained about 40% after one cycle of oscillation.” (lines 261-264);

“Damping rate of the $kaiA^-$ oscillation (0.057) is much higher than that of the $kaiC^{EE}$ rhythm (0.016).” (lines 291)

We believe these quantitative analyses much improved data validations, and thank the reviewer a lot.

1-2) Bioluminescence reporters for $P-kaiBC$ have been observed/studied before which led to an opposite conclusion in the past. This contrast has to be addressed more deeply, and better discussed.

We have clearly this point as follows:

“It should be noted that in previous studies, at least one^{23,24} or two cycles³ of P_{kaiBC} bioluminescence were observed retrospectively, although they were considered arrhythmia at that time because the amplitude of the damped oscillation in $kaiA^-$ strains was extremely low compared with that of the wild-type strain. In these studies, a partial segment of the upstream region of the $kaiBC$ gene (previously named D4)²¹ has been used as the $kaiBC$ promoter to drive bioluminescence because of its highly expressing level. The selection of this promoter unit might be beneficial to detect the damped oscillation profile with lower expression levels due to the lack of $kaiA$.” (lines 77-82)

Moreover, the detailed quantitative analysis mentioned as above strengthened the damped oscillation phenotype more deeply. In addition, we realized that the damped oscillation is not evidently observed at lower light intensity (15 $\mu\text{mol}/\text{m}^2/\text{s}$), as newly attached Figure S1. Thus, we also added the following text:

“In addition, light intensity seems also important to detect the damped oscillation, since at lower light intensity (15 $\mu\text{mol}/\text{m}^2/\text{s}$) the bioluminescence rhythm was more rapidly damped without showing the second peak of the rhythm (Supplementary Fig. 1).” (lines 82-85)

1-3) Following this line, in Figure S2: What is meant by "Because the second peaks were not observed reproducibly, period lengths were not calculated." Were the second peaks frequently not observed reproducibly? There is no mention of this in the main text and it would be appropriate to mention it.

According to the reviewer's suggestion, we added the following text in the main text:

“At 35°C, the oscillation in the $kaiA^-$ became much more dampened than that at 32°C. The average level of bioluminescence at 35°C was less than 10% of that at 30°C, and the second peaks of bioluminescence were not reproducibly observed (Supplementary Fig. 5). The tendency to lower bioluminescence level at relatively higher temperature has been observed in WT, i.e., the peak value at 38°C decreased by about 30% of that at 30°C³¹.” (lines 149-153)

2) further experimental proof demonstrating the TTFL mechanism, e.g. RT-PCR for $kaiBC$ mRNAs. Obviously, without a functional KaiA the protein oscillator is not able to perform circadian phosphorylation rhythms of KaiC. However, there is no further evidence that instead a TTFL is responsible for the observed damped oscillations. Either the $kaiBC$ transcript level or KaiB and KaiC protein levels are expected to oscillate in the KaiA-absent scenario which should be shown experimentally.

We understand the point. However, this is too demanding experimentally. In the wild type strain, the peak-to-trough ration of the robust P_{kaiBC} bioluminescence, $kaiBC$ mRNA and KaiC protein rhythms in LL are ~14, 6 and 2, respectively under our experimental conditions. In the $kaiA$ -null mutant, the peak-to-trough ratio of the bioluminescence is at best 2.3 even for the first peak (Fig. 1). Although linear regression is not evident, if this damping ratio is applied, we would need to detect mRNA and protein cycles with the peak-trough ratio of at best ~2.0 to ~1.3, respectively, with much lesser temporal resolution (every 4 h for mRNA/protein vs every 30 min for bioluminescence) and much higher experimental deviation.

Considering this, we actually did the northern and western experiments as follows. This figure is for the reviewer only. Panel a and b show bioluminescence profiles of the wild type and $kaiA^-$ strains in continuous liquid culture used for sampling. Panel c shows western profiles for KaiC. Panel d and e show combined data of bioluminescence (promoter activity, gray), $kaiBC$ mRNA by qPCR (cyan), and KaiC protein level (magenta). The profiles in WT are almost same as previously reported. Profiles of $kaiA^-$ strain may also support somewhat fluctuating $kaiBC$ and KaiC abundances somehow correlating with the bioluminescence profile. Thus, these data seem consistent with the idea of TTFL, while more detailed and multiple experiments would be necessary.

3) more extensive discussion including further references, e.g. minimal Kai-based systems e.g. by Schmelling *et al.* 2017; the concept of a damped oscillator adopted from e.g. the human circadian clock; TTFL models for *S. elongatus* e.g. by Hertel *et al.* 2010 and Zwicker *et al.* 2010. Further, the paragraph on KaiBC complex formation ends with the statement that this complex is important for generation and period determination of the damped oscillation in the absence of KaiA. In my opinion, the explanation for this is a little too brief. The observation is first and foremost rather obvious and therefore it should probably be supported by further discussion or even further experiments.

We appreciate the reviewer's insightful suggestions. Initially, we revised introduction to add more information on possible hourglass models in *Prochlorococcus* and *Rhodospseudomonas* and the possible minimal components suggested by Schmelling *et al.* as follows. This revision should be informative to readers for more understanding of the results.

“In *Synechococcus*, inactivation of *kaiA* dramatically reduces the magnitude of both KaiC phosphorylation and *kaiBC* expression¹. Thus, KaiA has been reported repetitively as an essential clock component in the cyanobacterial circadian system. Interestingly, the *kaiB* and *kaiC* genes are found not only in cyanobacteria but also in other proteobacteria and Archaea, while *kaiA* is only found in cyanobacteria. Detailed phylogenetic analysis by Dvornyk and colleagues (2003) suggested that *kaiA* is evolutionarily the youngest among the three genes⁹. Some marine cyanobacterial species such as *Prochlorococcus marinus* MED4 and PCC 9511 are known to lack *kaiA*. It has been proposed that in these species *kaiA* gene was lost after evolution of the intact *kaiABC* system¹⁰. Consistent with the proposed role of KaiA, *kaiA*-lacking species fail to exhibit oscillation under continuous conditions^{11, 12}, whereas they display diurnal variations in transcription and cell cycle control under light–dark (LD) cycles^{13, 14, 15}. Moreover, the diurnal but not free-running rhythm in nitrogen fixation has been reported in even non-cyanobacterial purple bacterium, *Rhodospseudomonas pulstris*, which harbors *kaiB* and *kaiC* homologues without *kaiA*¹⁶. Differing from KaiC in *Synechococcus*, KaiA is not essential for enhancing the basal auto-phosphorylation activity of the KaiC homologue in *Prochlorococcus* MED4 and *Rhodospseudomonas*^{12, 16}. In both species KaiC homologues undergoes phosphorylation in the light and dephosphorylation in the dark^{16, 17}. Based on these results, prior research discussed the possibility of a non-self-sustaining timing system in cyanobacterial and purple bacterial species lacking *kaiA*^{11, 12, 16, 17, 18}. More recently, a comparative bioinformatics study by Schmelling *et al.*¹⁹ reported a possible conserved gene network composed of *kaiB*, *kaiC*, *sasA*, *rpaA*, *rpaB*, *ldpA*, *cpmA* and *ircA* among cyanobacterial species, and proposed these genes as possible components of the prototypic hourglass-like timing system. There are a couple of possible mechanisms for the timing system other than a self-sustained oscillator from a mathematical viewpoint. One mechanism is the hourglass model, which can respond to periodical environments but does not exhibit any oscillations under constant conditions. The other possibility is damped oscillation, which can display oscillations under constant conditions, although its amplitude can decay exponentially. In both *Prochlorococcus* and *Rhodospseudomonas*, these possibilities have not been experimentally validated.” (Lines 32-54)

Then, for TTFL models, we largely modified the Conclusion as follows, including suggested information of Hertel and Zwicker's papers as follows:

“The TTFL process is likely involved in generating the damped oscillation, whereas KaiB-KaiC complex formation is, at least in part, important for the period determination process. Although the damped oscillation was less robust than

sustained oscillation and less temperature-compensated, it can maximally resonate to cyclic environments with a period of 1 day. After our finding that TTFL is not essential for circadian rhythms⁶, TTFL in the cyanobacterial circadian system has been proposed as a secondary loop to support robustness of the self-sustained post-translational oscillator^{37, 50, 51}, forming a coupled positive and negative feedback loops for proper timing⁵², and proper entrainment^{50, 53}. Our simulation suggests that the TTFL-related damped oscillator can resonate with the post-transcriptional Kai oscillator for robust timing. It also strongly suggests that the KaiB-KaiC-based damped oscillator in the absence of KaiA would be functional, at least as an hourglass timing mechanism. Our study should provide valuable insights into the survival strategies of cyanobacterial species lacking *kaiA* or other bacterial species harboring only *kaiB* and *kaiC* homologues without *kaiA*^{11, 16, 17, 54}. The finding also supports an evolutionary hypothesis that a proto-circadian system might evolve without KaiA as a KaiB-KaiC-based damped oscillatory (or hourglass) timing system that can resonate to environmental cycles, and after evolving KaiA, it was organized into the intact KaiABC-based sustained oscillator⁵⁵. The KaiC-binding, KaiB-like domain of SasA is only conserved in cyanobacterial species but not in other bacteria. Thus, transcriptional output mechanism from the KaiBC system would be different in cyanobacteria and other bacteria, such as *Rhodospseudomonas*^{19, 56}. Since *Prochlorococcus* MED4 is thought to have lost *kaiA* after evolution of the intact KaiABC, its TTFL situation would be more similar to the damped oscillatory system in the *Synechococcus kaiA* mutant strain. In any cases, it would be intriguing to address if the resonance effect shown in our study with external cycles of environmental cues with different period length are observed in *Prochlorococcus* MED4 and *Rhodospseudomonas*.” (lines 332-351)

For Kai complex formation, we added the following discussion:

“Based on these observations, we suggest that KaiB-KaiC complex formation is important for the generation and period determination of the damped oscillation in the absence of KaiA. A possible mechanism would be as follows. Upon light onset, the KaiC-SasA complex formation would initiate and accelerate phosphorylation of RpaA, thereby activating the transcription of *kaiBC*. Subsequent increase in KaiB and KaiC proteins would facilitate KaiB-KaiC complex formation, which in turn reduces the amount of SasA-KaiC complex by substituting SasA with KaiB. The resulting KaiB-KaiC complex binds to CikA, which attenuates RpaA-phosphorylation. Thus, the *kaiBC* transcription is reduced to close the TTFL. Although accumulation of KaiB would trigger transition from SasA-KaiC to KaiB-KaiC, it should be noted that the rate of KaiB-KaiC complex formation is slow and relatively insensitive to change in concentration of the proteins⁴⁹. Instead, the assembly rate of the complex is highly dependent on biochemical property of KaiC (possibly, ATPase activity in the CI domain)⁴⁴, which is altered by the above-mentioned mutations⁴⁸.” (lines 275-285)

For the concept of damped oscillator, our model section has already discussed it. However, according to another reviewer’s suggestion, we modified a conclusive text suggested from the simulation as follows.

“The amplitude of TTFL was determined by how strongly the self-sustained oscillator affected the damped oscillator. The oscillation amplitude also depended on the natural period of TTFL (Fig. 6i); i.e., the agreement of the natural period between these oscillators makes the amplitude of TTFL larger through resonance. This numerical result suggests that the cyanobacterial clock system should take advantage of the TTFL-including damped oscillation with a period of about 24 h to amplify the oscillation amplitude of the whole clock system via resonance. (lines 323-327).

We believe this rewording also addresses the point. On the other hand, we did not include the damped oscillation for mammals because we do not think it necessary for the main context of our report. Moreover, according to another reviewer’s comment, we added similarity and difference between the *kaiA*-less and *kaiC[EE]* mutant oscillations, which also deepen discussion on this section.

4) reproducible (and ideally open access e.g. protocols.io, and code for modeling at github or zenodo) description of experimental methods, e.g. bioluminescence measurements. Bioluminescence Assay is insufficiently described. What is the decanal solution for, at what wavelength was the OD measured, were the cells excluded that did not show a signal? It is not clear what the purpose of trend elimination is. The original publication of the bioluminescence assay does not describe comprehensively what and how luminescence is evaluated and measured which is key for the major findings of the manuscript. Further, the experiment on the entrained circadian clock is also not described in detail. Are the light pulses randomly selected after 2, 4 and 6 hours or are there additional reasons for this? The conclusion regarding higher sensitivity to dark stimuli is not described sufficiently.

We appreciate this comment. According to this suggestion, we largely added detailed info on Methods. For bioluminescence assays:

“After cultivation of each strain in BG-11 liquid medium under LL for 2-4 days, 10- μ l aliquots of diluted cells (corresponding to an optical density at 730 nm of approximately 1×10^{-5}) were inoculated onto BG-11 solid medium in 35-mm plates. After 5 days under LL, cells were synchronized to two 12-h:12-h LD cycles, and then bioluminescence was measured under LL at 50 μ mol photon \cdot m $^{-2}$ \cdot s $^{-1}$ in the presence of 1% decanal solution as a substrate in each plate as described previously⁵⁷. The bioluminescence values were measured with a photomultiplier tube H7360-01MOD (Hamamatsu Photonics KK, Japan) and normalized to the number of colonies per plate (counts per colony). Because of the low signal-to-noise ratio in the damped bioluminescence profiles in the *kaiA*-null background, if necessary, we also provided de-trended data to show damped oscillation property with the following formula to remove trends with *lag* (=10 h)” (lines 366-375)

For photic entrainment/resonance experiments:

“Cells were prepared and entrained as described above in “Bioluminescence Assay”. After entrainment to two LD cycles, the plates were placed in LL and the measurements of bioluminescence were started. Each plate was removed from the bioluminescence monitoring system transiently and transferred to the dark for 2, 4 or 6 h at hour 4, 8, 12, 16, 20, 24 or 28 in LL. After this dark acclimation, each plate was returned to LL on the bioluminescence monitoring system, and the bioluminescence assay was resumed. The durations of phase-shift was calculated by comparing dark-pulse-acclimated bioluminescence rhythm with that without dark pulses. It should be noted that phase information is not available for above-mentioned detrended data because the moving average method generates a delay. Therefore, we compared peak time of each bioluminescence rhythm (not detrended data but original traces). For resonance experiments, cells were synchronized to two LD cycles and the bioluminescence was monitored as described above. During the bioluminescence measurement, each plate was transiently removed from the monitoring system to the dark for 2 h four times with an interval of 16, 20, 24, 26, 30, or 32 h to administrate dark pulses with different external cycles (*T* cycle).” (lines 377-388)

For codes of modeling, we uploaded them on github:

“Codes of two mathematical models in Fig 6 are available at <https://github.com/hito1979/NatCommun2020>.” (line 418)

We also revised other experimental methods as well as possible on this occasion.

Further there are several minor concerns which should be considered as well:

- kaiA-, kaiA-less, and, kaiA- and ∂ kaiA should be used consistently throughout the manuscript, and might be explained more in detail

According to the reviewer’s comment, we consistently represented *kaiA*-null nomenclature. We added information as follows:

“As described previously¹, *kaiA*⁻ and *kaiB*⁻ refer to inactivation of the genes using an insertional stop codon in the ORF to inhibit translation, whereas Δ *kaiA*, Δ *kaiABC* and Δ *kaiBC* refer to complete deletion of the genes via replacement with an antibiotics resistance gene. *kaiA*-null strains refer to all of *kaiA*⁻ and Δ *kaiA* strains.” (lines 359-361)

- lines 89 - 94: very difficult to read and to understand

We agree to the comment. For the relationship between PRC and limit cycle model, we explained the interpretation more informatively, adding similar precious works in other model organisms, as follows:

“The negative correlation between the amplitude of the rhythm and the magnitude of phase shifting has been reported in circadian systems in mammals²⁷, *Arabidopsis*²⁸, and the *kaiC^{EE}* mutant of *Synechococcus*²⁶. These phenomena have been interpreted partly by a simple schematic model. Self-sustained circadian clocks have been considered to be limit cycle oscillators. In limit cycle theory, a reduction in the amplitude of the self-sustained oscillator is visualized as a limit cycle with a smaller diameter. A stimulus that causes an equivalent change in state variables would give rise to larger phase shifts if the trajectory of the damped oscillation is much smaller than the diameter of a high-amplitude limit cycle on the phase diagram. By contrast, the same stimulus could cause smaller phase shifts with a limit cycle with larger diameter²⁹. Thus, the difference in PRCs between the wild-type and *kaiA*⁻ strains could be discussed similarly: the wild-type and *kaiA*⁻ oscillations would be considered to be a self-sustained limit cycle with larger amplitude and the damped oscillator with smaller amplitude, respectively.” (lines 106-116)

- lines 95 - 97: *could be discussed more deeply*

According to this suggestion, we added a related reference and modified the text as follows:

“An alternative possibility is the enhancement of photic input pathways in *kaiA*⁻ strains. For example, disruption of glucose-1-phosphate adenylyltransferase gene (*glgC*) magnifies the dark-induced phase shifting through metabolic changes with a rapid fall of ATP/ (ATP + ADP) energy charge in dark conditions, while the amplitude of P_{*kaiBC*} rhythm is less altered³⁰. Although we cannot exclude this possibility, the former model appears more plausible because it is evident that the core oscillatory mechanism must be much more fragile in the absence of KaiA compared with the findings for the canonical circadian pacemaker in the wild-type strain.” (lines 116-121)

- lines 166 - 167: *please explain why more KaiBC interferes with the bioluminescence? Might be also important for data analysis.*

We modified the text as follows:

“The lower bioluminescence (*kaiBC* promoter activity) is most likely due to a negative feedback effect from overproduced KaiB and KaiC proteins under the control of D4 promoter activity.” (lines 215-217)

- lines 191 - 192: *I do not understand this sentence, please explain further.*

We corrected the text as follows for clarity:

“The result that the KaiC^{AA} mutant protein fails to show damped oscillation is interesting, because in the original *kaiA*⁻ strain the (wild type) KaiC is also hypophosphorylated but still able to produce the damped oscillation.” (lines 240-242)

Reviewers' Comments:

Reviewer #1:

Remarks to the Author:

The authors have addresses all my concerns. I think the resulting manuscript is clearer and makes an important contribution to the field.

Reviewer #2:

Remarks to the Author:

Dear authors,

Many thanks for taking all suggestions very seriously into account. I highly appreciate the new modeling approach. It clearly strengthens the message of the manuscript. Text and content of the study did improve a lot. Thus, I would recommend the manuscript for publishing.

Only a minor remark might remain. One could shortly explain to the readers that mRNA quantification of kaiBC via RT-PCR was not performed more intensively due to technical limitations, but will be an important addition in the future to further demonstrate the TTFL mechanism. Northern and Western "data seem consistent with the idea of TTFL"(Supplement Fig. X)", while more detailed and multiple experiments would be necessary."

REVIEWERS' COMMENTS:

Reviewer #1 (Remarks to the Author):

The authors have addressed all my concerns. I think the resulting manuscript is clearer and makes an important contribution to the field.

Reviewer #2 (Remarks to the Author):

Dear authors,

Many thanks for taking all suggestions very seriously into account. I highly appreciate the new modeling approach. It clearly strengthens the message of the manuscript. Text and content of the study did improve a lot. Thus, I would recommend the manuscript for publishing.

Only a minor remark might remain. One could shortly explain to the readers that mRNA quantification of *kaiBC* via RT-PCR was not performed more intensively due to technical limitations, but will be an important addition in the future to further demonstrate the TTFL mechanism. Northern and Western "data seem consistent with the idea of TTFL" (Supplement Fig. X)", while more detailed and multiple experiments would be necessary."

We are glad to hear the reviewer's opinion. For the minor remark, we added the preliminary data ($n=1$) as Supplementary Figure 10, which had been previously shown only to the reviewer, and mentioned in the main text as follows.

"It is experimentally difficult, however, to confirm that TTFL is at work by monitoring oscillating profiles in KaiBC protein or *kaiBC* mRNA abundance in *kaiA*⁻. In the wild type strain, the peak-to-trough ratios of the robust P_{*kaiBC*} bioluminescence, *kaiBC* mRNA and KaiC protein rhythms in LL are ~14, 6 and 2, respectively under our experimental conditions. In the *kaiA*-null mutant, the peak-to-trough ratio of the bioluminescence is at best 2.3 even for the first peak (Fig. 1). Although linear regression is not evident, if this damping ratio is applied, we would need to detect mRNA and protein cycles with the peak-trough ratio of at best ~1.6 and ~1.2, respectively, with much lesser temporal resolution (every 4 h for mRNA/protein vs every 30 min for bioluminescence) and much higher experimental deviation. Considering this difficulty, however, we examined it in the *kaiA*-null mutant. The profiles of bioluminescence, mRNA and KaiC in WT are almost same as previously reported (Supplementary figures 10d). The *kaiA*⁻ strain showed low-amplitude fluctuating *kaiBC* and KaiC abundances, as expected, somewhat correlating with the bioluminescence profile (Supplementary figures 10e). Thus, these preliminary data could be supportive for the idea of TTFL, while more detailed and multiple experiments are necessary." (lines 333-345)

Since the data were obtained by $n=1$ experiments (mentioned in the legend), however, it may be insufficient for the editorial policy standard, we concern (note that even though $n=1$, they are all time-sampled data with multiple time-points). If this is the case, we may remove this text and figure upon the editor's consideration. Even without this, we believe the manuscript makes sense.